# Multistep diversification in spatiotemporal bacterial-phage coevolution

Einat Shaer Tamar [1] & Roy Kishony [1,2,3] ✉

The evolutionary arms race between phages and bacteria, where bacteria evolve resistance to phages and phages retaliate with resistance-countering mutations, is a major driving force of molecular innovation and genetic diversification. Yet attempting to reproduce such ongoing retaliation dynamics in the lab has been challenging; laboratory coevolution experiments of phage and bacteria are typically performed in well-mixed environments and often lead to rapid stagnation with little genetic variability. Here, co-culturing motile *E. coli* with the lytic bacteriophage T7 on swimming plates, we observe complex spatiotemporal dynamics with multiple genetically diversifying adaptive cycles. Systematically quantifying over 10,000 resistance-infectivity phenotypes between evolved bacteria and phage isolates, we observe diversification into multiple coexisting ecotypes showing a complex interaction network with both host-range expansion and host-switch tradeoffs. Whole-genome sequencing of these evolved phage and bacterial isolates revealed a rich set of adaptive mutations in multiple genetic pathways including in genes not previously linked with phage-bacteria interactions. Synthetically reconstructing these new mutations, we discover phage-general and phage-specific resistance phenotypes as well as a strong synergy with the more classically known phage-resistance mutations. These results highlight the importance of spatial structure and migration for driving phage-bacteria coevolution, providing a concrete system for revealing new molecular mechanisms across diverse phage-bacterial systems.

Natural microbial communities harbor a broad range of species interacting within spatially heterogeneous environments such as soil, plants or the human gut. Within these complex communities, a key factor in species diversity is the antagonistic relationship between bacteria and their bacteriophage parasites[1–4]. When bacteria are challenged by lytic infection, selective advantage arises for phage-resistant bacteria, and consecutively, for phages that are able to overcome such evolved bacterial resistance[5–9]. These selective pressures, therefore, induce ongoing reciprocal adaptation that leads to the evolution of new resistance and counter-resistance pathways[10,11], contributing to the high spatiotemporal diversity of bacterial species and phage genomes in nature.

In striking contrast, when co-cultured in laboratory controlled well-mixed liquid environments phage-bacteria coevolution typically stagnates after just a few adaptive cycles[12–21], rarely showing ongoing coevolution dynamics[22–26]. Theory suggests that prolonged host-parasite coevolution and diversification may be facilitated by spatial structure[27–29], allowing local interactions and the formation of sub-populations. Indeed, introducing spatial structure into bacteria-phage co-evolution, by co-culturing them as colonies, in biofilms, in unshaked or small connected liquid environments, or in soil, has demonstrated more prolonged phage-bacteria coexistence and diversification[30–38]. Yet, a comprehensive long-term spatiotemporal

[1]Faculty of Biology, Technion–Israel Institute of Technology, Haifa, Israel. [2]Faculty of Computer Science, Technion–Israel Institute of Technology, Haifa, Israel. [3]Faculty of Biomedical Engineering, Technion–Israel Institute of Technology, Haifa, Israel. ✉e-mail: rkishony@technion.ac.il

tracking of naturally migrating bacteria and phage on a spatially structured laboratory environment has been missing. In particular, it is unknown whether a structured environment may facilitate co-evolution with more adaptive steps and higher degree of genetic diversification, potentially uncovering new genetic pathways.

The challenge of evolutionary stagnation and limited diversity in well-mixed environments is particularly exemplified in the laboratory coevolution dynamics of *Escherichia coli* and the lytic phage T7. Previous studies show that coevolution of *E. coli* and T7, or the closely related phage T3, in chemostats rapidly reach stagnation, typically after only 1.5 cycles (bacteria-phage-bacteria mutations)[12,13,15,21]. Both the phage and the bacteria show limited genetic diversity. In the phage, the observed genetic variations were limited to a single codon at the phage tail-fiber gene[15]. In the bacteria, mutations were focused at the lipopolysaccharides (LPS) biosynthesis genes[15,39] which also function as the phage receptor[40], the *trxA* gene[15] encoding for the phage DNA polymerase processivity factor[41], or the transcription regulator *rcsB* leading to colanic acid overproduction[21] forming a barrier to T7 infection[42]. This limited evolved diversity stands in contrast with the rich genetic pathways discovered in comprehensive genetic T7-resistance screens[42–44], most likely due to global interactions and to artificial population dilution which are inherent in well-mixed environments and impose strong selection for the fittest mutants.

Here, developing a spatiotemporal long-term bacterium-phage co-culturing assay, we discover multi-step co-retaliating adaptation paths leading to spatially-coexisting bacterial and phage phenotypic and genotypic diversification. Adapting a recently introduced swim-based setup, where chemotactic bacteria and lytic phages propagate on large semisolid agar plates[45,46], we co-evolve *E. coli* and their T7 bacteriophages over two one-week evolution rounds, observing continuous growth, migration and infection cycles, leading to evolutionary diversification and a mosaic spatial distribution of phage and bacteria. High-throughput mapping of interactions between evolved bacteria and phages reveals diverse resistance and infectivity classes, demonstrating host-range broadening and host switching dynamics. Diversification is further supported by multiple genetic variations, in both known and unknown *E. coli*-T7 interaction pathways, including a new resistance function in two bacterial genes. These findings stress the importance of a spatial component in facilitating diversification and genetic innovation during phage-bacteria coevolution.

## Results

### Swimming-plate time-lapse imaging reveals recurring spatio-temporal bacterial-phage co-evolution dynamics

We established a spatially structured coevolution platform by adapting swimming plates populated with motile bacteria and lytic phages for long term incubation. Following Ping et al.[45], we inoculated a mixture of bacteria and phages at the center of a swimming plate and monitored their growth dynamics with time-lapse imaging. Extending these earlier studies, we allowed coevolution on four plate replicates for a total of 15 days; Initially, bacteria and phages coevolved spontaneously for seven days (Fig. 1a, b, Supplementary Movie 1; Methods: Coevolution experiment), then small local samples from the seventh day were reinoculated on fresh swimming plates and coevolution continued for an additional eight days (Supplementary Fig. 1a, b; Supplementary Movie 2; Methods: Coevolution continual experiment). A main challenge in long-term incubation is water evaporation, which increases the effective agar concentration and obscures imaging due to condensation on the lid. To reduce evaporation, we sealed the plates with a heated glass lid, preventing evaporation and condensation[47]. To obtain high image resolution, plates were stored on custom-made dark-field LED illumination boxes (Supplementary Fig. 2), and time-lapse images were taken at intervals of 10 min., enabling highly accurate recording and analysis.

The obtained time-lapse videos showed recurrent waves of propagating bacteria followed by phage infection in all replicates (Supplementary Movie 1). Unlike in liquid environments, evolution did not end with bacterial dominance; analyzing changes in pixel intensities across space and time, we observed heterogeneous plate colonization with both organisms coexisting at separate spatial patches at the endpoint, as evident by areas of high final pixel intensity, representing bacterial growth, alongside areas of high difference between maximal and final intensity values, representing phage proliferation (Fig. 1c, green and magenta respectively). Counting the number of peaks in pixel intensity over time in small sites (1.4 mm wide squares) illustrated local growth and infection cycles. The median number of cycles on all replicates was 2; yet, several distinct locations showed a much higher number of growth-infection cycles (up to 12 cycles; Fig. 1d, e, Supplementary Fig. 3). In contrast, a control set of 4 bacteria-only, no-phage, swimming plates did not show multi-peak dynamics, confirming the role of the phage in these cycles (Methods: No-phage control; Supplementary Movie 3; Supplementary Fig. 3). While peak amplitude gradually decreased during initial coevolution (Fig. 1e), inoculation on continual plates led to renewed strong growth-infection cycles and coexistence lasting for 8 additional days (Supplementary Movie 2; Supplementary Fig. 1c–e; Supplementary Fig. 3), suggesting that the reduced amplitude towards the end of the initial coevolution round was driven by nutrient depletion rather than ecology-evolutionary stagnation. Taken together, unlike typical observations in well-mixed environments[12,13,21], *E. coli* and T7 on swimming plates manifested ongoing coevolution for 15 days with retaliating dynamics, with neither species globally prevailing at any time point.

### Interaction mapping of evolved bacterium-phage isolates reveals phenotypic diversification via both arms race and host switch dynamics

To uncover the evolved phenotypic diversification of bacteria and phage during coevolution on the plates, we sampled multiple sites from all plates at the end point of both the initial and the continual coevolution experiments (51 different sites: 29 from the initial and 22 from the continual experiment). From each sampled site, up to two bacterial clones and up to four phage clones were isolated (Methods: Single bacterial colonies isolation, Two-color plaque assay; Supplementary Fig. 4). To sample phage diversely, the phage clones from each sampled site were isolated on a lawn containing a mixture of differentially labeled wildtype bacteria and bacterial clones isolated from the same site (Supplementary Fig. 5). Overall a library of 94 bacterial isolates and 112 phage isolates were collected (Supplementary Data 1, 2). To assess the evolved resistance and infectivity phenotypes of all bacterial and phage isolates respectively, we measured all pairwise bacteria-phage interactions using a high-throughput plaque assay on omni-tray agar plates (total of 11,252 bacterium-phage interactions including 3 bacterial wildtype controls and 4 phage wildtype controls, Methods: Cross-infection assay). For each interaction, we automatically quantified an "infectivity score" reflecting the plaque size and turbidity (0 to 1, Methods: Automatic processing of plaque images; Supplementary Fig. 11). The robustness of the infectivity score was tested via measurement of correlation between all three bacterial wildtype infectivity scores and all four phage wildtype infectivity scores which showed a good agreement ($R^2 = 0.73–0.81$ between bacterial wildtype replicates, $R^2 = 0.99–1.00$ between phage wildtype replicates; Supplementary Fig. 6c, d). Finally, bacterial and phage isolates were clustered with hierarchical clustering into a comprehensive cross-infection matrix based on Euclidean pairwise distances of their infectivity scores (Methods: Dendrogram construction; Fig. 2a).

Systematically phenotyping the infectivity among bacterial and phage isolates, we identified multiple distinct adaptive phenotypes comprising a complex interaction network of bacteria and phages. Both bacteria and phage diversified into at least 9 and 12 distinct

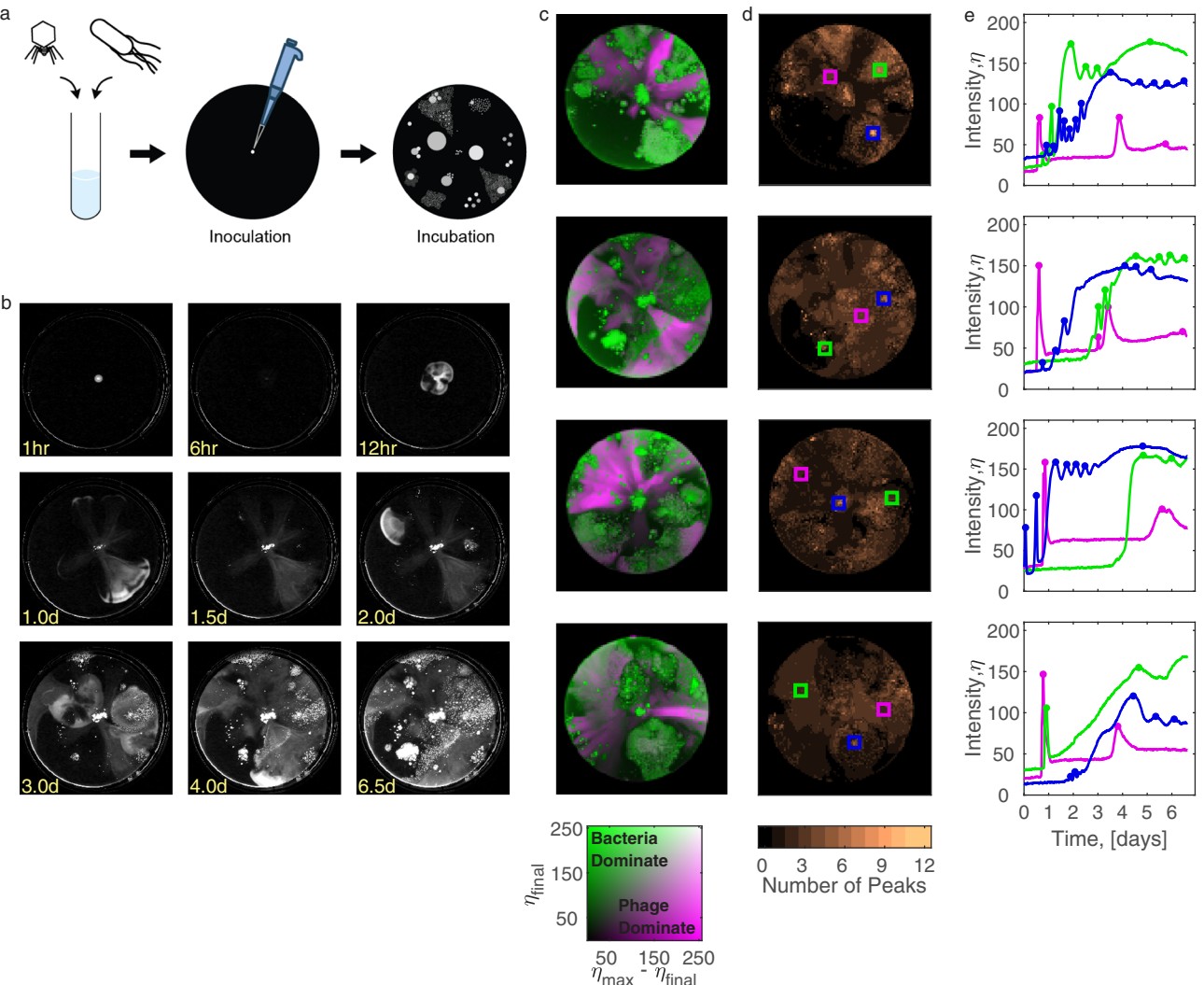

**Fig. 1 | Bacteria-phage coevolution on swimming-plates leads to multiple adaptive cycles with neither species taking over. a** Experimental setup for long-term evolution: mixtures of motile MG1655 cells and T7 virions are inoculated at the center of four replicate swimming plates (14 cm diameter, 70 ml of 0.3% agar in LB) and incubated at 30 °C for 7 days. **b** Sample dark-field images of a representative replicate at different time points. **c** A composite heatmap of final pixel intensity ($\eta_{final}$, green) and the difference between the maximal and final intensity ($\eta_{max}$-$\eta_{final}$, magenta), representing bacterial or phage dominance respectively on each of the four replicates at the final time point (2D legend, bottom). **d** Heatmap of the number of observed growth-lysis cycles, as quantified by enumerating intensity peaks in each location (averaged over zones of 1.4 mm width). **e** Intensity over time at specific locations (rectangles in **d**), chosen to represent areas of bacteria-dominance (green) or areas of phage-dominance (magenta) at the endpoint and highly dynamic areas with multiple intensity peaks (blue). The dots represent detected peaks. Source data are provided as a Source Data file.

phenotypic classes, respectively (Fig. 2a, dendrogram branch colors; Methods: Dendrogram construction). Notably, many of these classes evolved independently in different evolutionary replicates, indicating parallel co-evolution (Fig. 2a, dendrogram leaf colors). Some phenotypic classes, and in particular the most resistant bacterial classes and most broad ranged phage classes, were identified predominantly in the continual plate, supporting evolutionary arms-race. To further analyze the network structure of the evolved phenotypes we utilized the BiMat[48] library and found that, consistent with expectations for diverse and interconnected environments[49,50], evolved bacteria broadened their resistance profile and evolved phages extended their host range compared to their ancestors (all 4 coevolution plates showed significant nestedness, P < 10⁻⁴; only two of them also showed significant modularity, $P = 0.01$ and $P < 10^{-4}$, Methods: BiMat analysis; Supplementary Fig. 7). Yet, some phages underwent specialization by increasing infectivity against evolved bacteria while significantly reducing infectivity against the wildtype bacteria ("host switch", Fig. 2b). Comparing the infectivity scores to a more straightforward

infection measurement of plaque turbidity, similar phenotypic features were observed, including the host-switch phenomenon, however with lower capacity to differentiate between weaker infections and noise (Supplementary Fig. 6a, b; Methods: Automatic processing of plaque images). In all, our systematic phenotypic analysis revealed broad diversity in both resistance and infectivity classes showing signals of parallel arms-race evolution[51], with host-switch dynamics.

## A regression model identifies adaptive mutations in known and unknown phage-bacteria interaction pathways

At the genotypic level, association among mutations and infection phenotypes revealed genes and pathways important for adaptation of both bacteria and phage. Whole genome sequencing of the bacteria and phage isolate libraries revealed 40 SNPs, 49 indels and 12 amplifications in the bacterial isolates, and 192 SNPs and 18 indels in the phage isolates (Supplementary Data 3 and 4 respectively). Both bacteria and phage showed high ratios of nonsynonymous to synonymous mutations, indicating positive adaptive evolution (bacteria: 5

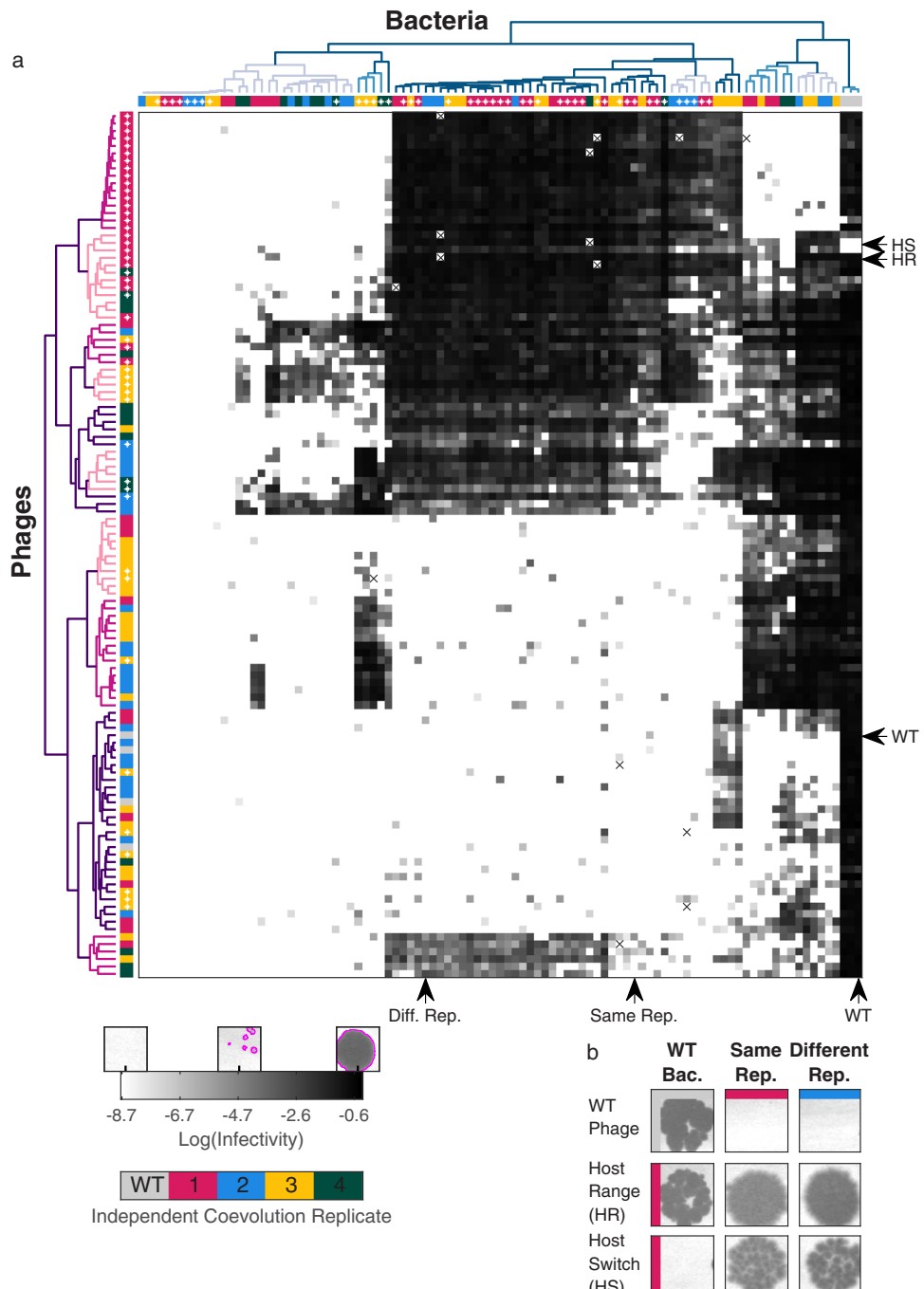

**Fig. 2 | Systematic profiling of interactions among evolved bacteria and phages reveals multiple distinct classes of coexisting phenotypes. a** Results of a cross-infection assay of all phage isolates (116, rows) on all bacterial isolates (97, columns). Squares in the matrix indicate the infectivity scores, calculated as the area showing phage lysis times the level of bacterial clearance in this area (Methods: Automatic processing of plaque images). Missing data points are marked with x. Isolates are ordered by hierarchical clustering based on phenotypic Euclidean pairwise distances, with each phenotypic class colored with a different color in the dendrograms (left and top branches). The color of each isolate (dendrogram leaves) represents one of the four independent co-evolution replicates and gray indicates the ancestral strain. White dots on the dendrogram leaves indicate isolates sampled from the continual coevolution plates. Source data are provided as a Source Data file. **b** Images of the infection assay for phage-bacteria isolates (arrows in **a**) demonstrating host-range expansion (HR) and host-switching (HS) phages. WT - wildtype, Diff. - Different, Bac. - Bacteria, Rep. - Replicate.

intergenic, 32 nonsynonymous, 3 synonymous, $P_{dN/dS} = 0.0078$; phage: 15 intergenic, 154 nonsynonymous, 23 synonymous, $P_{dN/dS} < 10^{-4}$; Methods: Sequencing analysis; Supplementary Fig. 8). To quantify the association of mutations with the infection phenotypes, we built a linear regression model for the infectivity score of each bacterium-phage pair (Fig. 2a) as a function of their joint genotypes. To avoid overfitting and resolve a compact set of mutations, we solve this

model using the Lasso technique (Least absolute shrinkage and selection operator, Methods)[52]. Bacterial mutations were strongly biased towards negative associations with infectivity, as expected for resistance mutations (Fig. 3a). Consistent with prior reports, these bacterial mutations contributing to resistance included mutations in LPS biosynthesis, Exopolysaccharides production (*lon*[53]), *trxA*, and the pyruvate dehydrogenase encoding *aceE*[54] that specifically binds to T7

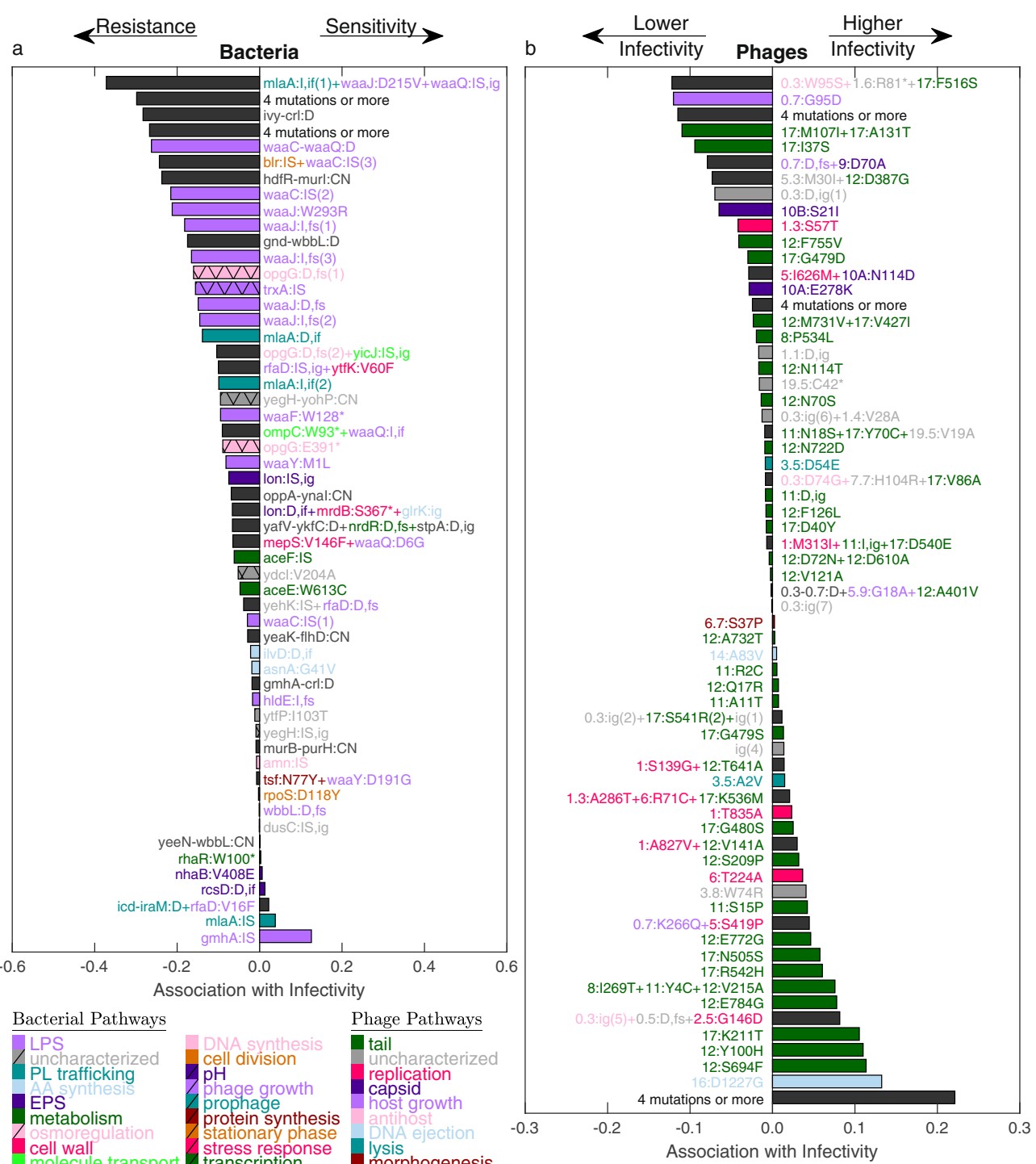

**Fig. 3 | Lasso regression analysis identifies mutations affecting phage-bacteria interactions.** Association of infectivity with mutations based on a joint regression model for all phage-bacteria interaction pairs, shown for all bacterial (**a**) and phage (**b**) mutations with non-zero Lasso coefficients. Bar colors represent the biological pathway of each mutated gene, with black bars indicating mutations affecting several genes of different biological pathways. AA - Amino acid, EPS - Exopolysaccharides, LPS - lipopolysaccharides, PL - phospholipid. Mutations are labeled according to their type: copy number variation (CN), deletion (D), insertion (I), insertion sequence (IS), intergenic (ig), inframe (if) and frame-shift (fs). Index (in parentheses) is added to different mutations with the same label. The numbers in the beginning of each phage mutation name (**b**) refer to the phage gene number ("gp" was omitted due to limited space). For example, 11:R2C represents a change in the second codon of the phage tail tubular gene *gp*11. Source data are provided as a Source Data file.

*gp0.4* with a yet unknown function[55]. Intriguingly, two additional genes that acquired several resistance mutations (mutations negatively associated with infectivity, Fig. 3a) were not previously linked with *E. coli* resistance to T7: the osmoregulated periplasmic glucans (OPG) biosynthesis protein *opgG* and the intermembrane phospholipid transporter *mlaA*. Focusing on *mlaA* mutations, while three out of four

mutations were assigned with a negative coefficient ("mlaA:I,if(1)", "mlaA:I,if(2)", "mlaA:D,if", Fig. 3a, Supplementary Data 3), one mutation received a positive coefficient ("mlaA: IS", Fig. 3a, Supplementary Data 3). This insertion sequence mutation caused a disruption of the *mlaA* gene, as opposed to the other three short insertion or deletion mutations which resulted in an in-frame addition or deletion of 2

codons after the 40th codon. Interestingly, the isolates carrying "mlaA:IS" were also carrying one of the short in-frame indels in *mlaA* (Bac33, Bac34, Supplementary Data 3), suggesting that the increased resistance caused by the in-frame mutation in *mlaA* was counteracted by the insertion sequence mutation that disrupted the gene. Similarly, the highest positive coefficient was assigned to an insertion sequence mutation in *gmhA* ("gmhA:IS", Fig. 3a) which is expected to cause a short LPS phenotype and increased T7 resistance on a wildtype background[42]. This mutation appeared in an isolate carrying an in-frame insertion in *mlaA* (Bac71, Supplementary Data 3), which shows increased sensitivity compared to other isolates carrying similar *mlaA* mutations and a wildtype *gmhA* (Supplementary Data 3 and Supplementary Data 5), as reflected in the positive association with infectivity. Since *gmhA* is essential for LPS biosynthesis, it is possible that the effect of the *mlaA* mutation on phospholipid transport was revoked by the deleterious effect of the mutated *gmhA* on LPS production.

In contrast with the bacterial mutations, phage mutations included many more mutations associated negatively with infectivity (Fig. 3b), presumably due to strong non-linearities such as the host-switch (Fig. 2b) and the ability to sample less fit phage genotypes which survived on the plate. Phage mutations included, as expected, mutations in the tail-fiber gene *gp17*, but also across the tail tubular genes *gp11* and *gp12* and even in non-structural genes including *gp16* and *gp1*, encoding for the DNA ejection protein and the RNA polymerase, respectively. A combination of 5 mutations highly associated with infectivity, including SNPs in *gp1* (T23A), *gp2* (D12G), *gp16* (I197T) and *gp11* (E164A) as well as an in-frame insertion sequence in *gp11*, were detected in a single phage isolate with a notably extended host range (Fig. 3b, mutation set with highest association with infectivity; Phg29 in Supplementary Data 4 and Supplementary Data 5). Finally, the same key set of genes, including phage tail genes, LPS and EPS biosynthesis genes, as well as the newly discovered *mlaA* and *opgG*, were also identified in a complementary parallel evolution analysis for genes mutated more than expected by chance (Methods, Multi-mutated genes), reinforcing these genes as driving adaptive evolution (Supplementary Fig. 9).

### Genetic reconstruction of mutations in *mlaA* and *opgG* suggests a new role of these genes in phage resistance

We confirmed the phage-resistance role of the newly identified *mlaA* and *opgG* mutations by synthetic reconstruction and competition-based phenotyping. Focusing on the *mlaA* gene, an ancestral strain transformed with a plasmid expressing a specific in-frame deletion observed in the evolution experiment (Fig. 3a, "mlaA:D,if", previously reported as a dominant gain-of-function mutation[56], had a mild but robust advantage over the wildtype bacterial strain when competed in the presence of the wildtype phage (Fig. 4a; dye swap in Supplementary Fig. 10a; Methods: Reconstruction of *mlaA* and *opgG* mutants, Two-color plaque assay). Furthermore, the actual independent resistance phenotype as measured individually could be even larger than observed in the two-color plaque assay experiment due to the existence of a sensitive strain that supports phage infection. Focusing on the *opgG* gene, synthetically reconstructing a nonsense mutation observed in the evolution experiment (Fig. 3a, "opgG:E391*"), we initially observed only a mild resistance advantage over the ancestral background against the wildtype phage (Fig. 4b, dye swap in Supplementary Fig. 10b), consistent with the absence of *opgG* in prior genetic screens for T7 resistance[42–44]. However, noticing that all the *opgG* mutated isolates also carried an additional mutation in the LPS glucosyltransferase *waaJ* gene, we then constructed a double mutant carrying both the *opgG* mutation and a *waaJ* mutation (*waaJ* W293R, the most frequently observed mutation in the co-evolution experiment). Notably, while this *waaJ* mutation alone also caused only a mild phenotype, the two mutations showed strong synergistic interaction; the *opgG* mutation had a strong resistance phenotype when added on

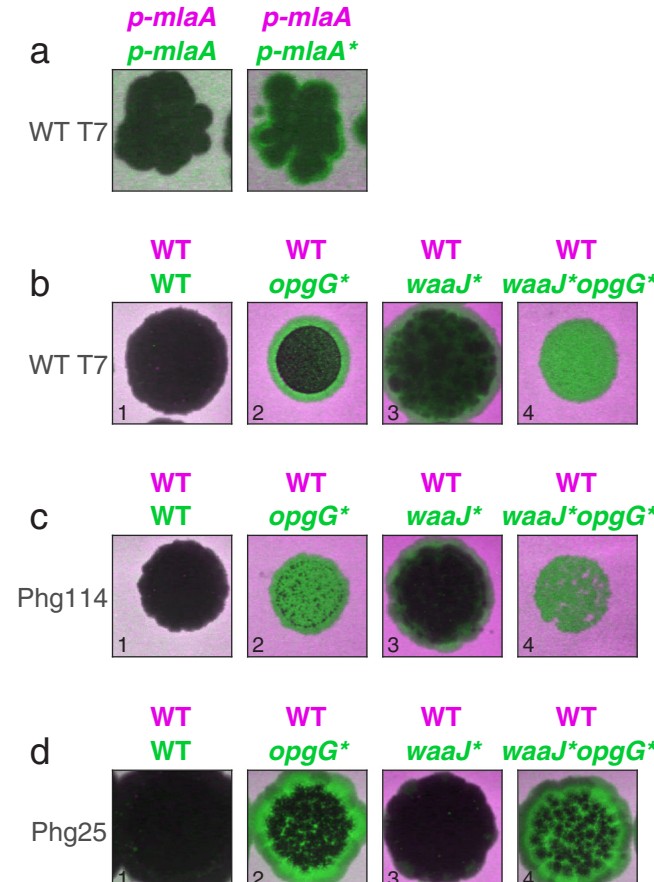

**Fig. 4 | Reconstructing mutations in *opgG* or *mlaA* provides T7 resistance alone and in interaction with a mutation in LPS biosynthesis. a** A two-color plaque assay with the wildtype T7 phage (WT T7) spotted on a mixture of YFP and mCherry tagged wildtype bacterial strains carrying a plasmid expressing either *mlaA* wild-type (p-*mlaA*) or *mlaA* mutant (p-*mlaA*\*) genes[56]. Left: a control with both strains expressing the wildtype *mlaA*; Right: a strain expressing *mlaA*\* and tagged YFP (green) has a small selective advantage over a strain expressing the wildtype *mlaA* (tagged mCherry, magenta), manifesting as a green halo at the border of the plaque zone (Methods: Two-color plaque assay). **b** An *opgG* mutant has a mild T7 resistance advantage over the wildtype (compare panel b2 versus wild-type control, panel b1) and strong synergy with the *waaJ* mutation (compare double mutant in panel b4 versus single mutants in panels b2, b3). **c, d** Unlike in resistance to the wildtype T7, when competed in the presence of two chosen evolved T7 phages (Phg114, **c** Phg25, **d** See genotypes in Supplementary Data 4 and infectivity phenotypes in Supplementary Data 5), the *opgG* mutation provides strong resistance both alone (c2, d2) and when combined with the *waaJ* mutation (c4, d4), indicating phage-specific advantage of this mutation.

a *waaJ* mutated background (Fig. 4b, dye swap in Supplementary Fig. 10b, WT T7). Furthermore, we also found that when facing some of the evolved phages, the *opgG* mutation alone was sufficient for a pronounced resistance phenotype (Fig. 4c, d, Supplementary Fig.10c, d). Overall, these results suggest a new role of *opgG* and *mlaA* in T7 resistance.

## Discussion

Using long-term coevolution on spatially structured environments, we revealed the diversification of *E. coli* and T7 into distinct resistance and infectivity classes through multiple steps of co-adaptation. In contrast to the often limited laboratory coevolution observed in well-mixed setups, in spatially structured swimming plates we observed ongoing growth and infection cycles which led to striking phenotypic variability, including initial steps towards phage speciation through host switch, in general a rare phenomenon in lab experiments[57–60] and

reported for T7 only in an artificially tailored system[61]. Variability was evident also at the genotypic level, where a Lasso-based phenotype-genotype association analysis revealed a wide range of resistance mechanisms, including two potentially novel T7 resistance mechanisms via mutations in *mlaA* and *opgG*. The resistance phenotype of *opgG* adds to related observations of endogenous-lysis resistance of OPG lacking *E. coli* cells[62], and of T7-resistance of transposon-mutagenized *opgH*, a gene residing on the same operon as *opgG*[44]. The resistance phenotype of the *mlaA* mutation may be related to a recently shown hyper-vesiculation phenotype of this mutation[56,63]. Outer-membrane vesicle secretion in general is known to increase phage resistance through irreversible binding of the phage to the vesicles[64,65], suggesting a similar resistance mechanism against T7. Evolved phages showed high genetic variability with more than 50 unique mutation events in the tail fiber gene *gp17* and more than 40 unique mutation events in the tail tubular gene *gp12*, many of which were associated with increased infectivity in the Lasso model. Mutations associated with infectivity were also detected in non-tail related genes such as the phage RNA polymerase encoding *gp1* and the *gp16* gene involved in DNA ejection.

Our results differ from coevolution outcomes in well-mixed setups, where studies, focusing on coevolution of *E. coli* and either the T7 or the closely related T3 phages reached stagnation after just 1.5 cycles (bacterial-phage-bacteria mutations)[12,13,15,21]; Parallel evolution was evident also at the genotypic level, with bacterial first resistance mutations in a single LPS biosynthesis gene, phage retaliating mutations in the same phage tail gene codon, and bacterial second resistance mutations either in additional LPS biosynthesis genes, in *trxA*[15], or in the capsule synthesis regulatory genes *rcsBC*[21]. Our study shows longer mutational paths and uncovers mutations in other genes and other pathways. It is likely that spatial structure plays a role in allowing higher genetic diversity, though other factors such as population size and overall evolutionary time may also play a role.

Our study has several limitations. First, sampling was done only at the end point of each coevolution round. Adding temporal sampling to our method, although challenging due to the risk of contamination, will allow a more detailed temporal reconstruction of the bacteria-phage co-phylogeny, revealing the chronologic adaptation steps and identifying bacteria-phage retaliating mutations. Second, our study relies on prior literature for comparison to well-mixed experiments. We note that setting up a perfectly matched well-mixed control is important, though both technically and conceptually challenging. A key challenge is matching population size over time as the mode of growth will diverge rapidly: in the well-mixed setup the bacteria grow exponentially and simultaneously, as opposed to growth on the swimming plates which is approximately quadratic, with fast growing bacteria at the front and slow growing bacteria at the back. Furthermore, the swimming plates differ from a well-mixed environment in two aspects: one is the spatial structure and the other is migration (as opposed to solid, non-swim, agar plates). It is not obvious how to properly set liquid experiments to resolve these two aspects. Finally, in this study we did not engage in genetic reconstruction of mutations in the phage, which will potentially reveal new phage gene functions and interactions between bacterial mutations and phage mutations.

We anticipate that expanding our method with longer evolution times and larger spatial scales can help reveal additional unexplored resistance and counter-resistance mechanisms in *E. coli* - T7 and in other bacterial-phage systems, possibly helping in the discovery of novel phage defense and anti-defense systems. Applied to clinically relevant pathogens and their phages, such experiments can also help predict co-adaptation steps to design durable phage therapy[66–70]. Overall, these results provide a general selection scheme, as well as

analysis methodologies, that can help reveal novel molecular mechanisms of phage-bacteria interactions.

## Methods

### Strains and growth conditions

Experiments were conducted with Bacteriophage T7 kindly provided by Debbie Lindell (Technion – Israel Institute of Technology), and a motile *Escherichia coli* MG1655 strain carrying IS1 upstream to *flhD* kindly provided by Ady Vaknin (The Hebrew University of Jerusalem) transformed with a plasmid constitutively expressing YFP, CFP or mCherry and a kanamycin resistance gene[71]. Cells were cultured in LB (10 g/L tryptone, 5 g/L yeast extract, 5 g/L sodium chloride in distilled water, autoclaved) supplemented with 30 μg/ml kanamycin at 30 °C with shaking unless otherwise noted. A T7 lysate was prepared by propagating 100 μl of the original T7 phage stock in 10 ml of exponentially growing MG1655 cells in LB (without antibiotics, $OD_{600} \approx 0.1$) until culture clearance was reached (75 min. at 30 °C). The lysate was sterilized with a 0.22 μm Millex-GV filter, supplemented with 1% chloroform and stored at 4 °C in a glass tube. Phage titers were estimated by plating serial dilutions of the lysate mixed with 100 μl MG1655 cells and 4 ml warm 0.7% agar on LB-agar plates (30 μg/ml kanamycin) and counting plaque forming units (PFUs) unless otherwise stated.

### Coevolution experiment

Swimming plates were prepared by pouring 70 ml of autoclaved semisolid medium (0.3% agar in LB, sterile distilled water was added after autoclaving to make up for the evaporated volume) supplemented with 30 μg/ml kanamycin into each of four polystyrene petri dishes (⌀ 14 cm). Plates were left to set for 1 h at room temperature and placed on custom-built dark-field imaging boxes equipped with LED light pads (Supplementary Fig. 2) positioned under a DSLR camera (Canon EOS 100D equipped with a Canon EF-S 18-55 mm f/3.5-5.6 IS II SLR Lens) in a temperature controlled room (30 °C, 70% humidity). Since long-term agar plates are susceptible to contamination, bacterial fluorescence markers (YFP,CFP) were used to allow visual detection of contamination. MG1655-CFP and MG1655-YFP overnight cultures were diluted 1/100 in LB (25 ml, 30 μg/ml kanamycin), incubated for 3 hours ($OD_{600} \approx 0.23$), centrifuged for 10 min at 2465 g and concentrated by resuspension in 190 μl LB. The concentrated cultures were sampled for CFU counting and mixed in a 1:1 volume ratio. However, in retrospect, the YFP marker has dominated already at the beginning, hence the information of marker ratio was not utilized and evolved MG1655-CFP isolates were excluded from subsequent analysis. Prior to inoculation of the plates, 30 μl of cells were mixed with 30 μl of a $10^{-6}$ diluted T7 lysate, and 5 μl of the mixed culture ($6 \cdot 10^6 \pm 1 \cdot 10^6$ MG1655-YFP cells, $8 \cdot 10^5 \pm 3 \cdot 10^5$ MG1655-CFP cells, and $13 \pm 2$ T7 virions) was inoculated at the center of each plate. To avoid condensation on the lids, plates were covered with a pre-warmed electrically heated glass (85 cm X 65 cm, Seaclear Industries LLC) connected to a power supply (OFI Electronics Ltd) set to 15 V and 1.48 A. The temperature of the glass was monitored and ranged from 30-32 °C throughout the experiment. Images were captured automatically every 10 min (Canon, EOS utility, exposure = 0.5 s, aperture F = 10, ISO = 100, white balance = daylight) during 7 days (a total of 954 images).

### Sampling

Following 7 days of incubation, 5−8 chosen areas from each plate were sampled (Supplementary Fig. 4) by gentle local mixing of the soft agar with a tip and pipetting out 5 μl into 245 μl phosphate buffered saline (pH 7.4, Sigma-Aldrich P-5368, one pouch was dissolved in one liter of distilled water and autoclaved). 140 μl of each sample were mixed with 70 μl glycerol 50% (final concentration 16.7%) and stored at −80 °C. To obtain a separated phage sample, 100 μl from each sample were mixed with 11 μl chloroform (~10%) and stored in small glass tubes at 4 °C.

## Coevolution continual experiment

Four new swimming plates were prepared as described above. Four chosen samples from the initial experiment, one from each replicate (Magenta samples in Supplementary Fig. 4), were thawed and inoculated (10 µl of the glycerol stock) at the plate centers. Incubation and imaging continued as described above for a total of 8 days (1153 images), whereafter between 2 and 12 areas were sampled from each plate as described above (Supplementary Fig. 4).

## No-phage control

Four new swimming plates were prepared as described above. A MG1655-YFP overnight culture was diluted 1/100 in LB (25 ml, 30 µg/ml kanamycin), incubated for 3 h, centrifuged for 10 min at 2465 g and concentrated by resuspension in 190 µl LB. 15 µl of the concentrated culture were mixed with 45 µl LB (to replace the T7 and MG1655-CFP culture volume in the coevolution assay) and 5 µl were inoculated at the center of the four swimming plates (prepared as described under "Coevolution experiment"). The plates were placed on the dark-field imaging setup (Supplementary Fig. 2) and covered with a heated glass (~32 °C). Plates were incubated (30 °C, 70% humidity) for 33 h and imaged every 10 min. as described above.

## Image analysis

Images were analyzed with custom MATLAB scripts. The first image of each experiment (initial coevolution / continual coevolution / no-phage control) was removed from all subsequent images to reduce background noise, the images were cropped around the plates, and only the red channel which showed the highest signal-to-noise ratio was taken. Time lapse videos of both coevolution rounds were constructed with LRTimelapse 5.6.0 (24 FPS and ¼ speed) from 300 (initial experiment) or 340 (continual experiment) images with logarithmically increasing time gaps. Time lapse movie of the no-phage control was constructed with Adobe Premiere Pro 2022 and contained all 200 images with constant time gaps. To calculate organism dominance, pixel values were averaged over areas of $10 \times 10$ px$^2$ and for peak analysis values were further averaged on a sliding time window of four images (30 min). Peak analysis was done with the MATLAB findpeaks function (MinPeakDistance = 20, MinPeakProminence = 4).

## Single bacterial colonies isolation

A total of 51 sampled areas were chosen: 29 from the end of the initial coevolution round and 22 from the end of the continual evolution round (Supplementary Fig. 4). To isolate single colonies from each area, the frozen samples were thawed and 10 µl of each sample were streaked on LB-agar (30 µg/ml kanamycin). After 2 days of incubation, up to two single colonies from each plate were picked and propagated in 2 ml deep-wells (400 ul LB, 30 µg/ml kanamycin) overnight. Stocks (16% glycerol) were maintained at −80 °C. A total of 94 colonies were isolated for further analysis. While colonies were not restreaked to remove possible carryover, the following cross-infection experiment (see below) was designed to sensitively detect any possible carryover: each omni-tray was overlaid with a culture of a single bacterial isolate and spotted with 96 drops, with around 25% of the drops not containing any phage isolate by design. In case of phage carryover, phage plaques were expected to appear in some of the empty drops or in the surrounding area. However, analysis of data showed no indication of carryover in any of the isolates.

## Two-color plaque assay (single plaque isolation)

Since T7 phages can persist for several days without replicating, we suspected that phage samples can contain, in addition to the evolved phages, wildtype phages from earlier time points. In order to isolate a diverse set of evolved phages, we established a fluorescence-based plaque assay: All bacterial isolates from both coevolution rounds (YFP) and a wildtype MG1655 (mCherry) were grown overnight, and a mixture of 60 µl of each isolate and the wildtype in 4 ml agar (0.7%) was overlaid on agar plates. Plates were divided into 6 equal segments and 10 µl spots of 1/10 serial dilutions of the corresponding phage samples (phages and bacteria from the same sampled area) and the wildtype T7 phage were spotted in the outer and inner parts of each segment, respectively (example in Supplementary Fig. 5). After incubation, plates were imaged with YFP and mCherry filters using a Macroscope (macro-scale fluorescence imaging device[72]). Phages that could only infect the wildtype (mCherry) strain formed green plaques (surviving YFP isolates), phages that can infect both strains formed black plaques, and phages that infect the evolved strain (YFP) but lost infectivity against the wildtype (mCherry) formed magenta plaques (host-switch, Supplementary Fig. 5). While the colors of the plaques were not directly processed in our analysis, this step gave us, in addition to plaque isolation, a first glimpse into the evolved phenotypes and the minimal number of evolution cycles by testing both the evolved resistance of the bacterial isolate to the wildtype phage and the evolved infectivity of the phage isolate against the bacterial isolate. Up to 2 plaques from each plate, preferably with different color and morphology, were picked and inoculated into exponentially growing bacterial culture of either the wildtype or the evolved strain from the same sample, depending on the plaque color, in order to avoid additional adaptation cycles during phage propagation; Green plaques were propagated with wildtype MG1655, and black or magenta plaques were propagated on the bacterial isolate from the same phage sample. Following 2 h of incubation with slow shaking, phages were transferred into 96-Well Sample Prep plates with tapered 0.7 mL glass inserts (Analytical Sales and Services), supplemented with 10% Chloroform and stored at 4 °C. A total of 112 plaques were isolated for further analysis.

## Cross-infection assay

As a preliminary step, the PFU of each phage isolate and the ancestral T7 was measured by a standard PFU assay over the permissive host strain (wildtype or evolved), and all isolates were standardized to a fixed titer of ~$3.3 \cdot 10^4$ PFU/ml. Next, all bacterial isolates and the wildtype MG1655 were grown overnight. Then, 120 µl of each isolate were mixed in 7 ml agar (0.7%), overlaid on one-well LB-Agar plates and inoculated with ~4 µl spots of each phage isolates (100-150 PFUs) using a PLATEMASTER (Gilson). Plates were incubated for 6.5 hr at 30 °C for growth and infection and imaged with the Macroscope.

## Automatic processing of plaque images

Cross-infection assay images were processed with MATLAB custom scripts. Briefly, each plate was divided into 96 squared regions centered at the phage spots ($L \times L = 181 \times 181$ px$^2$, there are 96 non-overlapping squares in each plate). Each square was normalized so that all pixels above the 90 percentile in the square are equal to 255; Then, the infectivity of each phage isolate against each bacterial isolate was calculated within the corresponding square as follows:

$$\text{Infectivity} = \text{phageFraction} \times \text{clearance} \tag{1}$$

Where phageFraction is the relative area of phage lysis in a given region,

$$\text{phageFraction} = \frac{\text{phageArea}}{L \times L} \tag{2}$$

calculated by a predefined plate-specific threshold (with region-specific thresholds for outliers), and clearance is defined as:

$$\text{clearance} = 1 - \text{mean}\left(V_{\text{norm}}^{\text{isPhage}}\right), \tag{3}$$

where $V_{norm}^{isPhage} \equiv \frac{V(isPhage)}{V_{max}}$ are all pixel values $V$ within the phage area in a given region, divided by the maximal pixel value $V_{max} = 255$. Their mean is calculated by summing all $V_{norm}^{isPhage}$ values and dividing by the phage lysis area.

To test the validity of the infection score, a more straightforward turbidity score was calculated by taking the sum of all pixel values, $V$, in a given phage spot square from the normalized plates, and dividing by the area of the square $L \times L$ and the maximum pixel value $V_{max}$:

$$\text{Turbidity} = \frac{1}{V_{max} \times L \times L} \sum V \tag{4}$$

Turbidity score values are expected to act inversely to infectivity scores: a plaque with a high infectivity score should receive a low turbidity score and vice versa.

## Dendrogram construction

Bacterial and phage isolates were clustered according to their resistance and infectivity phenotypes, based on their infectivity score vectors. Specifically, the Euclidean pairwise distances between all bacterial isolates were calculated with the MATLAB pdist function. These distances were clustered with the MATLAB linkage function for agglomerative hierarchical cluster trees (method = 'complete'). Finally, the cross-infection matrix columns were ordered and the dendrogram was visualized with the MATLAB dendrogram function. Isolates were divided to classes with the ColorThreshold option based on a threshold of 40% of the maximum linkage. The same process was repeated for the phage isolate clustering and ordering of the cross-infection matrix rows.

## BiMat analysis

We applied the MATLAB BiMat package[48] to calculate the nestedness and modularity properties of each replicate infection matrix including the ancestors. Missing data values (15 in total) were assigned with the mean of their $5 \times 5$ surrounding squares in the class-sorted matrix (Fig. 2) and all values were binarized. Following Gupta et al.[73], modularity and nestedness values were calculated with the default settings while ignoring empty rows and columns. Statistical significance was calculated as the chance to get equal or larger modularity or nestedness values among $10^4$ random matrices constructed with the EQUI-PROBABLE null model.

## DNA sequencing

Bacterial genomic DNA was extracted by centrifuging 400 μl of fresh bacterial cultures and resuspending the pellets in Lysis buffer (20 mM Tris, 2 mM EDTA, 1.5% Triton, pH 8.0) supplemented with 7 mg/ml lysozyme and incubating for 40 min in a 37 °C water bath. DNA was isolated with the DNA, RNA, and protein purification NucleoSpin® TriPrep kit (MACHEREY-NAGEL). Treatment with proteinase-K was carried for 30 min at 56 °C. Phage genomic DNA was extracted by incubating 400 μl of fresh lysates with proteinase-K (75 μg/ml final) and SDS (0.5% final) for 75 min at 56 °C, then isolating DNA with the DNA, RNA, and protein purification NucleoSpin® TriPrep kit (MACHEREY-NAGEL). Bacterial and Phage DNA concentrations were measured with a High Sensitivity Quant-iT™ dsDNA Assay Kit (Invitrogen) and standardized to 1.5 ng/μl and 1 ng/μl respectively. Sequencing libraries were prepared according to Baym et al.[74]. and sequenced with Illumina HiSeq X machine to produce 150 base paired-end reads (Admera Health). Raw sequencing files were deposited to the SRA database, see bioProject PRJNA884167 for phage sequencing data and bioProject PRJNA884682 for bacteria sequencing data.

## Sequencing analysis

Illumina reads were filtered to remove reads contaminated by the Nextera adapter or low-quality bases (>2 bases with a Phred Score of <20). SNP analysis was performed using an in house analysis pipeline[75]. Briefly, reads were aligned to a reference genome (Genbank U00096.3 for bacteria and NC_001604 for phages) using Bowtie 1.2.1.1 with a maximum of 3 mismatches per read. Base calling was done using SAMtools and BCFtools 0.1.19, and a genome position was determined as a SNP when more than a single allele was identified among all isolates with a quality threshold of FQ < −80. In parallel, insertions and deletions were identified with Breseq version 0.32.0[76] with the same reference genomes. To identify amplifications and additional deletions, the genome coverage of each isolate was normalized in two steps: first, the coverage values of each isolate were divided by the isolate median coverage, and then the normalized values were further divided by the corresponding normalized base pair coverage value of the ancestral genome. Areas with high or low coverage were identified manually. Adaptive evolution significance ($P_{dN/dS}$) was evaluated by randomly distributing the total number of intragenic SNPs across all ORFs X10,000 times, while accounting for mutations that appeared in more than one replicate and preserving transition/transversion ratios.

## Lasso regression analysis

We combined the genotypic and phenotypic data of all bacterial and phage isolates into a regression model to identify the mutations with the strongest positive or negative effect on infectivity. Specifically, the infectivity score of each phage-bacteria isolate pair (Fig. 2, Supplementary Data 5) was used as the response variable in the regression model, with the predictors being the joint phage and bacterial mutations of the corresponding bacterial isolate and phage isolate. SNP and deletions were assigned a binary value and amplifications were assigned an integer according to their estimated copy number. Long amplifications and deletion events that overlapped between isolates were segmented to overlapping and non-overlapping regions. For example, a deletion from 1000 bp to 3000 bp in isolate A and a deletion from 2000bp to 4000 bp in isolate B will be truncated into 3 unique events: a deletion between 1000 bp and 2000 bp in isolate A, a deletion between 2000bp-3000bp in both isolates, and a deletion between 3000 bp and 4000 bp in isolate B. Mutations that always appeared together were joined, except for amplifications (due to the difference in the assigned predictor values which were dependent on copy number), and synonymous SNPs were omitted, leading to a total of 204 mutations. Taking only unique genotypes and the average phenotype of pairs with identical genotypes, we ended up with 5636 observations. Lasso regularization was conducted with the MATLAB built-in Lasso function using X5 cross validation and max lambda = 200. Chosen mutation coefficients were within one standard error of the minimum mean squared error.

## Multiply-mutated genes

In order to detect genes that were mutated more than expected by chance, we simulated a random mutation distribution as follows: the entire bacterial or phage mutation set was taken with all information except for the mutation position in the reference genome. For each simulation round, random positions were drawn from the reference genome for each mutation and the affected genes were documented (long amplification and deletion mutations may affect multiple genes). If a mutation appeared in more than one replicate (parallel evolution), its position was drawn as the number of replicates in which it appeared. We ran the simulation 500 times for bacterial mutations and 5000 for phage mutations and calculated the distribution of the number of mutations per gene. The multi-mutated gene threshold was set where only 5% of the simulations reached the experimental value.

## Reconstruction of *mlaA* and *opgG* mutants

*OpgG* mutants were constructed with MAGE[77] (Multiplex Automated Genome Engineering). MG1655 cells were transformed with pORT-MAGE311B (Addgene number 120418), and electrocompetent cells

were prepared following Szili et al.[78]. The electrocompetent cells (40 μL) were mixed with 2 μl of MAGE oligos (500 μM) in 1 mm cuvettes and electroporated with a 1.8 kV pulse, then suspended in 5 ml prewarmed Terrific Broth (yeast extract 24 g/L, tryptone 12 g/L, $K_2HPO_4$ 9.4 g/L, $KH_2PO_4$ 2 g/L in water) and recovered for 2.5 h at 37 °C under shaking. Finally, 100 μl of cells were plated on LB agar plates with kanamycin. *waaJ* mutants were first engineered and validated via both T7 resistance and MASC-PCR[79] (QIAGEN Multiplex PCR Kit with a 2 μM 1:1 primers mix). Then, both the *waaJ* mutated and the wildtype strain were engineered with the *opgG* mutation and colonies were validated with MASC-PCR. Finally, all strains were transformed with a YFP/mCherry expressing plasmids carrying a chloramphenicol resistance gene. All oligos and primers are listed in Supplementary Table 1, MAGE oligos were designed using the MODEST design tool[80]. *mlaA* mutants were constructed by transforming MG1655 cells with either pGS100-MlaA (wild type *mlaA*) or pGS100-MlaA102 (*mlaA* with a deletion of the 41st and 42nd codons[56]) kindly provided by Paola Sperandeo (University of Milan), together with a YFP/mCherry expressing plasmids via heat shock transformation. Overnight cultured of transformed cells were diluted 1/100 in LB supplemented with chloramphenicol (12.5 μg/ml), kanamycin (15 μg/ml) and IPTG (0.1 mM) and cultured for 4 h at 30 °C before plating, which was done according to the Two-color plaque assay on omni-tray plates.

### Reporting summary

Further information on research design is available in the Nature Portfolio Reporting Summary linked to this article.

### Data availability

Genome sequencing data is deposited in the public SRA database, see accession code PRJNA884167 for phage sequencing data and accession code PRJNA884682 for bacterial sequencing data. All other datasets generated and analyzed during the current study were deposited in the Zenodo repository with https://doi.org/10.5281/zenodo.7347986 [https://zenodo.org/record/7347986#.Y4aQbXbMLIU]. Source data are provided with this paper.

### Code availability

Code for data analysis and figure generation is available on GitHub (https://github.com/Technion-Kishony-lab/Phage-bacteria-spatial-interactions.git).

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

## Acknowledgements

We thank D. Lindell, A. Vaknin and P. Sperandeo for providing strains. We are grateful to I. Yelin for support with sequencing and experimentation. We thank B. Csörgő and Á. Nyerges for advice on the MAGE protocol and A. Tamar for advice on the Lasso analysis. We thank D. Russ, O. Milman, V. Lazar, C. Velling, A. Eldar and N. Aframian for thorough reading of the manuscript and valuable comments. This research was supported in part by the ISRAEL SCIENCE FOUNDATION (grant No. 455/19), by the ISRAEL SCIENCE FOUNDATION – BROAD INSTITUTE Joint Program, (grant No. 2790/19) and through an award from the Kavli Microbiome Ideas Challenge, a project led by the American Society for Microbiology in partnership with the American Chemical Society and the American Physical Society and supported by The Kavli Foundation (to R.K.).

## Author contributions

E.S.T and R.K. designed the study. E.S.T. performed experiments and analysis. E.S.T., and R.K. interpreted the results and wrote the manuscript.

## Competing interests

The authors declare no competing interests.
