## [Peer Review File · Nature Communications]

Multistep diversification in spatiotemporal bacterial-phage coevolutionReviewers' Comments:

Reviewer #1:

Remarks to the Author:

This paper describes dynamics of phage-bacterium interaction in a structured environment. This research brings forward an important aspect of laboratory studies in contrast to nature: typically phage-bacterium coevolution studies are done in liquid culture, whereas in nature the environment is more variable, allowing more diverse outcomes and the co-existence of both the phage and its hosts. Here, the authors have used an elegant plate co-culture approach to visually map spatiotemporal phage-bacterium dynamics. In line with expectations, co-evolution led to increased resistance in the bacterial hosts and wider host range in the phage. Interestingly, similar infectivity patterns evolved in the independent replicates, and mutations were identified by genomic sequencing of both phage and the host. A new type of resistance mutations were identified and confirmed by construction of mutant strains.

The experiments are well done, methods are state of the art, and the observations are interesting. Especially the dark field imaging and two-colour plaque assays are novel approaches.

My key issues with the MS are the superficial discussion and the complex way the methods and results are written – the reader needs to go back and forth with these two chapters to understand what has been done. For example, figure 2 has a lot of information to digest, which could be explained better in the text: it is not clear where the isolates originate from (see specific comments below).

Furthermore, I suggest the authors provide a more in-depth discussion of the results, considering also the perspective of phage evolution.

Specific comments

“Phage killing” : phage infectivity (or host range) is a more generally used term. Also “phage killing area”: plaque or lysis

Figure 1 + Supplementary figure 1. For transparency: were the representative rectangles selected after the data was collected?

Figure 2. A central question here is whether the data contains samples from both phases of the experiment (as indicated in results, lines 129-131), or just from either one. If both: what isolates originate from the first part of the coevolution study, and which ones from the second part? If only one phase: please indicate which one. How the phenotypic dendrogram was constructed has not been mentioned in the text.

Figure 3. Is phage infectivity score calculated here over all bacterial hosts? Please indicate what the codes in the b panel (phage) mean – probably ORF number and mutation coordinate? What are the 4 (or more) mutations that lead to highest infectivity/host range? Are they found together in the same phage genome? Please elaborate this in discussion.

Lines 291- Setting up the coevolution experiment is a bit unclear – what is the role of the MG1655-CFP bacteria in this experiment?

Line 332-335: single bacterial colonies from the second experiment? Or from both phases? Sampling is not mentioned for the second phase plates, but in the results (lines 129-131) it is mentioned that the isolates originate from both phases. Colonies were not purified to remove possible carryover of phages from the initial experiment? Furthermore, number of isolated colonies and phage plaques should be mentioned in methods.

Lines 437-441: How were the simulations for mutations performed?

Lines 443- this reviewer is not familiar with MAGE, what does this acronym mean and are there any references for the method?

Legends of the Supplementary tables 1 -5 are not sufficiently descriptive. E.g. What are the numbers provided in suppl table 1 and 2?

Phage and bacterial genomes should be submitted to GenBank

Reviewer #2:

Remarks to the Author:

The authors of the manuscript Multistep diversification in spatiotemporal bacterial-phage coevolution address why laboratory coevolution experiments between *E. coli* and phage T7 yield limited genetic diversity. Their experiments investigate how spatiotemporal separation via bacterial migration affects coevolution. The experimental design very nicely chooses a natural (i.e., not experimenter-induced) migration scheme by using swimming bacterial cells. They investigate the evolutionary outcome phenotypically using a killing assay and genotypically through whole-genome sequencing. Finally, a subset of mutations is investigated by genetic reconstruction.

The paper has been a pleasure to read, the experimental design and analysis are innovative and well thought-through. While we (the report has been prepared collaboratively following journal guidelines) are looking forward to seeing this work published, we would like to see two issues addressed: Firstly, the manuscript is arguably too concise in places, and we were left wondering about control experiments or in lieu thereof a more comprehensive discussion section. Secondly, while it is easy to read, it is not accessible in parts: Some techniques are just glanced over and presumably inaccessible to the broad readership of Nature Communications. In addition, the colour scheme chosen makes large parts of the paper inaccessible to colleagues who are colour blind. (We are not colour blind and believe could have interpreted all data.)

****Major concerns – scientific content:****

* Control experiments: While we commend the authors for the overall experimental design, we are somehow missing two control experiments. Firstly, a liquid control; literature is cited and can serve as a control, but we wonder how different plate reader results would be from Figure 1e). Secondly, a control without coevolution. To which extent would bacteria and phage evolve their relationship in the absence of the pathogen and presence of ancestor host, respectively? We do not suggest the authors perform and analyse all those experiments but are calling for an in-depth discussion.

Figure 1:

* We struggle to understand how robust the measure of 'bacteria dominated' versus 'phage dominated' is. In panel (e, first and second from top), yellow and green lines appear qualitatively very similar, but are classified as different.

* While we do not doubt the claim of continued coevolution, it would be very helpful to see more quantification. Firstly, a histogram of the number of peaks, and, secondly, an analysis that takes 'prominence' of the peaks (compare to topographic prominence) into account.

* Chemotaxis will also result in at least one peak. Could the authors elaborate on the null expectation of just swimming *E. coli* (pixel intensity reduction is possible due to bacterial migration)?

Figure 2 and corresponding text:

* Classification and statistical analysis are presented extremely concisely, even the supplementary material is brief (How was classification done? Why using the BiMat package?) and can barely be

assessed by the reader. Not only, but especially given the strong claim in the abstract, a more thorough discussion should be warranted.

Figure 3 and corresponding text:

* The authors explain the 'more killing' side for phages, but don't comment on the 'higher sensitivity' side for bacteria. We are intrigued by the contribution to sensitivity by *gmhA:IS*. (Knockout of gene *gmhA* provides resistance to T7 (Qimron, PNAS, 2006).)

* To our understanding, mutations are combined if they always occur together. Could the authors elaborate to avoid confusion with testing for epistasis? And why did this not apply to amplifications?

* One concern with pooling all observations of bacterium-phage pairs may be that mutations may have different effects depending on the genotype of the antagonist, potentially cancelling each other out in the analysis. Is this a valid concern? How did the authors address it?

The authors left out CFP cells from analysis. To our understanding, they obtained 90 % of expected CFP counts. Is that correct? Was it much less at the end? What was the rationale for leaving those cells out instead of using them as an additional data point to visualize the retaliatory dynamics possible between bacteria and phage, but at a different host growth rate?

The two-colour assay is a really nice technique. We are worried, though, about possible unintended effects. For example, a drop with virions of T7 *with high PFU count* can induce a plaque on a lawn of *trxA* knockouts. We hypothesise that the two-colour assay would show clearing for both WT and Δ *trxA* with WT T7. This would be because phage amplifies on WT leading to infection and lysis of Δ *trxA*, while those alone would not support plaque growth.

****Major concerns – presentation:****

We suspect Figure 1 is almost unreadable for colleagues with red-green colour blindness. We suggest the authors choose a colour scheme that is widely acknowledged to be accessible. Online tools are readily available to also test accessibility of more complex figures like Figure 3.

****Minor concerns:****

For reproducibility, could the authors share the design of the custom-made dark-field LED illumination boxes?

Are the results robust against a redefinition of the killing score, e.g., just taking turbidity and not area into account. Or just plaque growth (which might not be possible given one data point). A more intuitive explanation might be helpful.

Cross-infection assay: Were pictures normalised so that $V_{max}=255$ in all cases?

In Figure 2, the shading of the dendrogram is almost invisible and quantification of the colour bar on killing is missing.

Could the authors clarify whether phenotypic (and genotypic) analysis was done solely on samples after round 2 of the evolution experiment?

End of section on regression model: Reference to Methods would be helpful in last sentence.

Single plaque isolation: What is the purpose of wildtype T7 dilutions?

The authors did not engage in the study of the temporal aspect of co-evolution, since they chose to take samples only after the end of the co-evolution rather than sample from the same location at different timepoints, which we agree would be highly ambitious and prone to several technical

challenges. A discussion would be interesting, however.

Sentence "... E. coli and T7 on swimming plates manifested ongoing coevolution for 15 days with symmetric dynamics, with neither species globally prevailing at any time point.": We suggest replacing the term symmetric by one reflecting the inherent asymmetry of the system.

* Because of the number of free parameters, testing for epistasis is probably out of reach. Would it make sense, though, to see whether the epistatic effect between mutations in *opgG* and *waaJ* appears in the Lasso regression?

* A reference for MAGE and MASC-PCR might be helpful.

* In the caption Supplementary Figure 5, is the reference to Supplementary Figure 5 intended?

* In the caption of Supplementary Figure 6, do the numbers match up with the figure? Both plaques 1 and 2 appear green to us.

Reviewer #3:

Remarks to the Author:

Tamar and Kishony carried out experimental evolution of *E. coli* and T7 phage on four swim-agar plates that were imaged continuously and characterized bouts of coevolutionary dynamics as they occurred across space and time. They were able to observe multi-step coevolutionary cycles in distinct regions of space, with their own relatively uncoupled (?) dynamics [question mark here because actually I'm not sure this is something directly measured, but perhaps could be easily ascertained with the image data already on hand].

Tamar and Kishony further characterized infectivity between 116 phage isolates and 97 bacteria clones and identified canonical patterns observed in coevolutionary diversification. Further genetic analysis identified many mutations of possible adaptive benefit in both host and phage, but most interestingly mutations that seem to have reduced infectivity of the phage. Rather than clear arms-race dynamics, the phage evolved a host-switch, where adaptation to the new (T7wt) Resistant host genotype came at a cost to infectivity against the naive host.

This study will certainly be broadly interesting (and the movies are just stunning). The results are in line with observations seen in other bacteria-phage coevolution work, though it is rare to see all of these in a single study. The length and number of coevolutionary "bouts" is perhaps the most striking, which then enables the further analyses performed.

I only have a few comments, none of which should prohibit publication.

1) The introduction reads quite abrupt and disconnected. Many times I'm left either wanting a sentence to connect the main points of the paragraph with the current study. For example, in paragraph 2 (L445 - 53), the introduction of a system of spatiotemporal tracking is introduced as a gap in the literature, rather than as a way of addressing a particular question. Similarly, at the end of paragraph 3, I'm left wondering why coevolution experiments have failed to identify so many of the identified pathways to T7 resistance; a preview could really set the reader up for some of the interesting results to come!

2) [Minor] Line 38: The phrasing "subjected to lytic infection" read awkwardly to me, since we aren't the ones doing the infection per se.

3) [Minor] Line 41: I would eliminate the word "cycle", since reciprocal adaptation is itself a cycle

4) [Minor] Line 76: What is meant by "wide genetic variations"?

5) Lines 105-107: The median cycle being 2 is interesting, I wonder what the phenotypic outcome of these frequent 2-cycle bouts of coevolution are. Are they often the same set of adaptations in the 2-cycles, or are the endpoints of the 2-cycles found within longer cycles of coevolution and they're somehow cut short? What does the distribution of number of cycles look like?

6) L127-148: Since there are replicate infectivity analyses for the phage and bacteria wildtypes, I'd like to know how repeatable the automated scoring of infectivity was.

7) L165-193: One of the big differences between this study and other studies of host-parasite coevolution is that genotypes persist in the cultures without being removed. Does that mean that isolating spatial and temporal effects are more challenging? Or that dN/dS rates are potentially biased? If I'm right that the accumulation of previously fit genotypes that have since become disfavored remain in the culture and are sampled, I'd just like some mention of it.

Point-by-point response to reviewer comments

Reviewer #1 (Remarks to the Author):

This paper describes dynamics of phage-bacterium interaction in a structured environment. This research brings forward an important aspect of laboratory studies in contrast to nature: typically phage-bacterium coevolution studies are done in liquid culture, whereas in nature the environment is more variable, allowing more diverse outcomes and the co-existence of both the phage and its hosts. Here, the authors have used an elegant plate co-culture approach to visually map spatiotemporal phage-bacterium dynamics. In line with expectations, co-evolution led to increased resistance in the bacterial hosts and wider host range in the phage. Interestingly, similar infectivity patterns evolved in the independent replicates, and mutations were identified by genomic sequencing of both phage and the host. A new type of resistance mutations were identified and confirmed by construction of mutant strains.

The experiments are well done, methods are state of the art, and the observations are interesting. Especially the dark field imaging and two-colour plaque assays are novel approaches.

We thank the reviewer for appreciating the novelty of the approach and the significance of our findings, and are thankful for the highly constructive comments.

My key issues with the MS are the superficial discussion and the complex way the methods and results are written – the reader needs to go back and forth with these two chapters to understand what has been done. For example, figure 2 has a lot of information to digest, which could be explained better in the text: it is not clear where the isolates originate from (see specific comments below).

Following the reviewer comment, we have now expanded the Method section to be fully self contained. We have also added a new paragraph before Figure 2 that explicitly describes the experimental setup and the origin of the strains (see new paragraph “**High-throughput cross-infection assay for interaction mapping**”). We have also added a new figure that depicts the strain collection scheme and location on the plates (see new **Supplementary Figure 4**) and a new figure showing the design of the dark-field illumination setup (see new **Supplementary Figure 2**).

Furthermore, I suggest the authors provide a more in-depth discussion of the results, considering also the perspective of phage evolution.

Following the reviewer's suggestion we have now added in the discussion a section discussing the evolution and genetic variability in the phages (see **expanded Discussion**).

Specific comments

“Phage killing” : phage infectivity (or host range) is a more generally used term. Also “phage killing area”: plaque or lysis

We now replaced “killing” with “infectivity” throughout the text and figures. Also, the previous term “killing score” is now termed “Infectivity score”.

Figure 1 + Supplementary figure 1. For transparency: were the representative rectangles selected after the data was collected?

We have now revised the caption of Figure 1 and Supplementary Figure 1 to clarify that the rectangles were manually selected to represent areas of bacteria-dominated, phage-dominated, or highly dynamic areas with multiple intensity peaks.

Figure 2. A central question here is whether the data contains samples from both phases of the experiment (as indicated in results, lines 129-131), or just from either one. If both: what isolates originate from the first part of the coevolution study, and which ones from the second part? If only one phase: please indicate which one.

We revised Figure 2 to indicate for each isolate whether it came from the initial or continual coevolution round (**revised Figure 2**) and added a sentence relating to this newly added information, indicating that the most resistant and infective classes were dominantly found on the continual plates (**see lines 164-166**). We’ve also have added a new figure that shows the sampling scheme and origin of each isolate on the coevolution plates (see **new Supplementary Figure 4**).

How the phenotypic dendrogram was constructed has not been mentioned in the text.

We added a new section describing the dendrogram construction in the Methods (**new section: “Dendrogram construction”**).

Figure 3. Is phage infectivity score calculated here over all bacterial hosts?

We now better clarify that the same phage infectivity score which was calculated in Figure 2 for each phage-bacteria isolate pair is used in the Lasso analysis of Figure 3 as the response variable in the regression model, with the predictors being the joint mutations of the corresponding bacterial isolate and phage isolate (**see revised and expanded “Lasso regression analysis“ in Methods**).

Please indicate what the codes in the b panel (phage) mean – probably ORF number and mutation coordinate?

Indeed, we now clarify that the phage mutation code refers to the phage gene name and the mutation loci within the ORF. Caption has been revised to indicate that we omitted the “gp” before gene names due to limited space in the figure (**see revised caption of Figure 3b**).

What are the 4 (or more) mutations that lead to highest infectivity/host range? Are they found together in the same phage genome? Please elaborate this in discussion.

We've added a sentence explaining the combination of mutations with the highest association with infectivity, which were identified in a single phage isolate (**see paragraph 9 line 229-233**).

Lines 291- Setting up the coevolution experiment is a bit unclear – what is the role of the MG1655-CFP bacteria in this experiment?

These long-term agar plates are somewhat susceptible to contamination. Fluorescence markers were used to allow visual detection of contamination (no sign of contamination was found). Two markers were used to be able to detect genetic sweeps. However, in retrospect, the YFP marker has dominated already at the beginning and the information of marker ratio was thereby not utilized. **We added an explanation in Methods (under “Coevolution experiment“)**.

Line 332-335: single bacterial colonies from the second experiment? Or from both phases? Sampling is not mentioned for the second phase plates, but in the results (lines 129-131) it is mentioned that the isolates originate from both phases.

The same isolation procedure was repeated after both first and second evolution experiments and isolated from both rounds were included in the analysis. Following the reviewer comment, we have now rephrased the Methods to clarify that isolates were sampled from the continual plates as well, see **Methods lines 387, 413-414, 431**.

Colonies were not purified to remove possible carryover of phages from the initial experiment? Furthermore, number of isolated colonies and phage plaques should be mentioned in methods.

Due to the large-scale nature of these experiments, colonies were not restreaked to remove possible carryover. Yet, the experiment was designed to sensitively detect any possible carryover. In particular, in the cross infection assay each omni-tray was overlaid with a culture of a single bacterial isolate and spotted with 96 drops, with around 25% of the drops not containing any phage isolate by design . In case of phage carryover, phage plaques were expected to appear in some of the empty drops or in the surrounding area. However, analysis of data showed no indication of carryover in any of the isolates (**see Methods, revised Single bacterial colonies isolation**). The number of isolated colonies and plaques is now mentioned in Methods (**lines 417, 452**).

Lines 437-441: How were the simulations for mutations performed?

We now added a detailed explanation of the simulation procedure (**Methods, revised “Multiply-mutated genes”**).

Lines 443- this reviewer is not familiar with MAGE, what does this acronym mean and are there any references for the method?

We added a reference for the MAGE (multiplex automated genome engineering) technique which was indeed missing. We also resolve the acronym meaning in the text (**Methods, line 573**).

Legends of the Supplementary tables 1 -5 are not sufficiently descriptive. E.g. What are the numbers provided in suppl table 1 and 2?

We extended the Supplementary table legends to include detailed description of the information presented in each table.

Phage and bacterial genomes should be submitted to GenBank

Phage and bacterial raw sequencing data are now deposited in the SRA database. Accession numbers are provided (**Methods, “DNA Sequencing”**)

Reviewer #2 (Remarks to the Author):

The authors of the manuscript Multistep diversification in spatiotemporal bacterial-phage coevolution address why laboratory coevolution experiments between *E. coli* and phage T7 yield limited genetic diversity. Their experiments investigate how spatiotemporal separation via bacterial migration affects coevolution. The experimental design very nicely chooses a natural (i.e., not experimenter-induced) migration scheme by using swimming bacterial cells. They investigate the evolutionary outcome phenotypically using a killing assay and genotypically through whole-genome sequencing. Finally, a subset of mutations is investigated by genetic reconstruction.

The paper has been a pleasure to read, the experimental design and analysis are innovative and well thought-through.

We thank the reviewer for the thorough reading and appreciation of the methodological advances. We are in particular grateful for the highly constructive suggestions and comments.

While we (the report has been prepared collaboratively following journal guidelines) are looking forward to seeing this work published, we would like to see two issues addressed:

Firstly, the manuscript is arguably too concise in places, and we were left wondering about control experiments or in lieu thereof a more comprehensive discussion section. Secondly, while it is easy to read, it is not accessible in parts: Some techniques are just glanced over and presumably inaccessible to the broad readership of Nature Communications.

We have now expanded the Method section to make it more self-contained (in particular, we have added the following new sections: **Dendrogram construction**, **Automatic processing of plaque images**, and **No-phage control** and expanded all other sections). We have also extended the main text and in particular the description of methodology and results. As suggested, we've also added a discussion of the well-mixed controls (see more specific response below).

In addition, the colour scheme chosen makes large parts of the paper inaccessible to colleagues who are colour blind. (We are not colour blind and believe could have interpreted all data.)

Following the reviewer comment, we changed all red-green figures to magenta-green (see **revised Figures 1 and 4**, **Supplementary Figures 1,5,10**) and chose a different, color-blind friendly palette for **Figure 3**. We verified the accessibility of all figures to color blind users with the following simulator: <http://www.color-blindness.com/coblis-color-blindness-simulator/>

****Major concerns – scientific content:****

* Control experiments: While we commend the authors for the overall experimental design, we are somehow missing two control experiments. Firstly, a liquid control; literature is cited and can serve as a control, but we wonder how different plate reader results would be from Figure 1e).

This is indeed a very important comment which we have considered in much depth when designing the experiments. In general, these control experiments are not only technically challenging, but also difficult to define conceptually. A key factor in evolution dynamics is of course population size, and the two experiments have to somehow match in that respect. Yet, even if we succeed to match the initial population size in a well-mixed setup (for example, by simply inoculating the same number of bacteria that we use on the plate into liquid LB with the same volume (70 ml LB) that we have in the agar plates), the population size will diverge rapidly: in the well-mixed setup the bacteria will grow exponentially with all cells replicating until exhausting the nutrients and reaching stationary phase, as opposed to growth on the swimming plates which is approximately quadratic in rate (disk expansion), and with fast replicating bacteria on the front and staganting bacteria at the center. Adding the phages to these dynamics makes it even harder to match population size. Furthermore, the swimming plates differ from a well-mixed environment in two aspects: one is the spatial structure and the other is the migration (as opposed to solid, non-swim, agar plates). It is not obvious how to properly set liquid experiments to resolve these two differences. Finally, if we consider 96-well plates as a liquid control, we again face a technical challenge in controlling for population size. A reasonable liquid control would have contained an initial population concentration which is

similar to the inoculated bacterial and phage population on the plates. In a standard 96-well plate of 150ul per well (~15ml), this would mean ~10⁶ bacterial cells but just 3 phages for an entire plate, so that only 3 wells are likely to be inoculated with a phage particle. Finally, another technical challenge is the periodic dilution in a well-mixed environment, which we could not perform with the swimming plates. We have added a discussion of these issues in the revised Discussion section (**see revised discussion, lines 313-321**)

Secondly, a control without coevolution. To which extent would bacteria and phage evolve their relationship in the absence of the pathogen and presence of ancestor host, respectively? We do not suggest the authors perform and analyse all those experiments but are calling for an in-depth discussion.

Following the reviewer comment, we have now conducted a no-phage control experiment which confirmed that simple growth-saturation dynamics without observation of newly emerging lineages (**new Supplementary Movie 3**). A no-bacterial-evolution control is not feasible, technically, since the bacteria cannot be removed from the swimming plates without completely disrupting the spatial structure.

Figure 1:

* We struggle to understand how robust the measure of 'bacteria dominated' versus 'phage dominated' is. In panel (e, first and second from top), yellow and green lines appear qualitatively very similar, but are classified as different.

Following the reviewer comment we have now revised the caption of figure 1 and Supplementary figure 1 to clarify that the rectangles in panel e were manually selected to represent areas of bacteria-dominated, phage-dominated, or highly dynamic areas with multiple intensity peaks. We also changed some of the example intensity lines to ones better exemplifying phage/ bacterial dominance (**revised Figure 1 and caption**)

* While we do not doubt the claim of continued coevolution, it would be very helpful to see more quantification. Firstly, a histogram of the number of peaks, and, secondly, an analysis that takes 'prominence' of the peaks (compare to topographic prominence) into account.

Following the reviewer comment we now show the distribution of the number of peaks and the distribution of peak prominence during both rounds of coevolution, compared to our new experiment of bacteria propagating on swimming plates without phages (**new Supplementary Figure 3**). The peak analysis in Figure 1 (and Supplementary Figure 1 and 3 for the no-phage control) is based on minimal peak prominence (using the MATLAB findpeaks function). The function parameters are now added in Methods (**see "Image analysis"**)

* Chemotaxis will also result in at least one peak. Could the authors elaborate on the null expectation of just swimming E. coli (pixel intensity reduction is possible due to bacterial migration)?

Since the chemotaxis behavior on a swimming plate varies upon different *E. coli* strains and growth media, in order to address this comment we ran a new experiment with our bacterial strain propagating alone on swimming plates under the same growth conditions and imaging parameters of the coevolution assay (**Methods, “no-phage control” and new Supplementary Movie 3**) and analyzed peak distribution with the exact same parameters. We found that since the chemotaxis peak in the propagating bacterial front is very weak, no peaks are typically detected in these control experiments (the median number of peaks without the phage is 0; **see distribution of number of peaks and peak prominence in new Supplementary Figure 3**).

Figure 2 and corresponding text:

* Classification and statistical analysis are presented extremely concisely, even the supplementary material is brief (How was classification done? Why using the BiMat package?) and can barely be assessed by the reader. Not only, but especially given the strong claim in the abstract, a more thorough discussion should be warranted.

Following the reviewer comment we have now added a paragraph before Figure 2 to explain in more detail the methods and analysis (**Paragraph “High-throughput cross-infection assay for interaction mapping”**)

Figure 3 and corresponding text:

* The authors explain the ‘more killing’ side for phages, but don’t comment on the ‘higher sensitivity’ side for bacteria. We are intrigued by the contribution to sensitivity by *gmhA*:IS. (Knockout of gene *gmhA* provides resistance to T7 (Qimron, PNAS, 2006).)

Following this interesting observation, we now added a sentence addressing the higher sensitivity and specifically the insertion sequence in *gmhA* in the text (**see lines 215-223**). Briefly, this mutation appears on an isolate with a mutation in *miaA*, but is more sensitive than isolates carrying a mutation in *miaA* but not in *gmhA*, and we speculate that the increased resistance gained by the *miaA* mutation, presumably through increased phospholipids and secretion of OMV, is canceled by the mutation in *gmhA* that blocks LPS production.

* To our understanding, mutations are combined if they always occur together. Could the authors elaborate to avoid confusion with testing for epistasis?

Indeed mutations were combined in the regression model if they were always identified together, since we could not differentiate between their singular effect. The existence of such coupled mutations may indicate epistasis, but may also be related to sampling just at the end point, and analyzing a limited number of isolates. Hence we did not engage in testing for epistasis in the regression model, and further addressed specific mutation combination only when they appeared in different combinations (**see *miaA* indel mutations with *gmhA* insertion mutation, newly added lines 215-223 and *waaJ* mutation with or without mutation in *opgG*, lines 263-270**)

And why did this not apply to amplifications?

Following the reviewer's question, we now clarify that amplifications were treated differently in the model and were not combined when appeared with other mutations because their assigned model predictor values were related to their copy number, as opposed to 0/1 in the case of all other mutation types (**Methods, revised "Lasso regression analysis"**).

* One concern with pooling all observations of bacterium-phage pairs may be that mutations may have different effects depending on the genotype of the antagonist, potentially cancelling each other out in the analysis. Is this a valid concern? How did the authors address it?

Indeed we now better explain that the interaction between mutations is not linear, such as in the example of the insertion in *gmhA* (**Lines 215-223**)

The authors left out CFP cells from analysis. To our understanding, they obtained 90 % of expected CFP counts. Is that correct? Was it much less at the end? What was the rationale for leaving those cells out instead of using them as an additional data point to visualize the retaliatory dynamics possible between bacteria and phage, but at a different host growth rate?

The CFP count actually showed only 10% of the expected CFU count. Moreover, when randomly sampling the plates at the end of the initial coevolution round, less than 10% of the colonies originated from the MG1655-CFP strain. The rationale behind using fluorescence markers was to allow visual detection of contamination (no contamination was actually observed). However, in retrospect, the YFP marker has dominated already at the beginning (CFP CFU counts actually showed only 10% of the expected CFU count), and sampling at the end of the initial coevolution round yielded only 4 CFP colonies. Thereby the information of marker ratio was thereby not utilized. **We added an explanation in the Methods (under "Coevolution experiment")**

The two-colour assay is a really nice technique. We are worried, though, about possible unintended effects. For example, a drop with virions of T7 *with high PFU count* can induce a plaque on a lawn of *trxA* knockouts. We hypothesise that the two-colour assay would show clearing for both WT and *DtrxA* with WT T7. This would be because phage amplifies on WT leading to infection and lysis of *DtrxA*, while those alone would not support plaque growth.

We very much agree and we now stress in the text that the actual independent phenotypes as measured individually could be larger than observed in the two-color co-cultures experiment (**lines 257-259**).

Major concerns – presentation.

We suspect Figure 1 is almost unreadable for colleagues with red-green colour blindness. We suggest the authors choose a colour scheme that is widely acknowledged to be accessible. Online tools are readily available to also test accessibility of more complex figures like Figure 3.

Thank you for this comment. We changed all red-green figures to magenta-green (**Fig. 1, Supp. Fig. 1, Fig. 4, Supp. Fig. 5 and 10**) and chose a different, color-blind friendly palette for Figure 3. We further tested the accessibility of all figures to color blind users with the following simulator: <http://www.color-blindness.com/coblis-color-blindness-simulator/>

****Minor concerns:****

For reproducibility, could the authors share the design of the custom-made dark-field LED illumination boxes?

We added a figure with the design of the dark-field illumination setup for swimming plates (**new Supplementary Figure 2**).

Are the results robust against a redefinition of the killing score, e.g., just taking turbidity and not area into account. Or just plaque growth (which might not be possible given one data point). A more intuitive explanation might be helpful.

Following the reviewer comments, we tested the robustness of the infectivity score by comparing it to a more straightforward turbidity score by directly taking the sum of all pixels in a predefined plate area around the plaque center which is equal to all samples (**new Supplementary Figure 7**). We observe generally similar matrix features and the host switch phenomena is evident. The turbidity scores are noisier in the lower infectivity range (high turbidity) compared to the low infectivity values which are calculated only within real plaques (**lines 173-176 in main text and lines 481-486 in Methods**).

Cross-infection assay: Were pictures normalised so that $V_{max}=255$ in all cases?

Pictures were indeed normalized, we have added an explanation of the normalization procedure (**Methods, "Cross-infection assay"**).

In Figure 2, the shading of the dendrogram is almost invisible and quantification of the colour bar on killing is missing.

Following this comment we changed the dendrogram branch colors and enlarged the branch width in Figure 2 and added numbers to the infection score colorbar.

Could the authors clarify whether phenotypic (and genotypic) analysis was done solely on samples after round 2 of the evolution experiment?

The entire analysis was done on samples from both rounds of evolution experiment, and this is now stated clearly in the revised text and in Figure 2. We have added a new figure that shows the sampling scheme and origin of each isolate on the coevolution plates (see **new Supplementary Figure 4**).

End of section on regression model: Reference to Methods would be helpful in last sentence.

A reference to Methods was added.

Single plaque isolation: What is the purpose of wildtype T7 dilutions?

While we haven't integrated the infectivity of the wildtype T7 over the isolated bacteria in our analysis, this step gave us, in addition to plaque isolation, a first glimpse into the evolved phenotypes and the number of evolution cycles by testing both the evolved resistance of the bacterial isolate to the wildtype phage (if plaques formed by the wildtype T7 were green) and the evolved infectivity of the phage isolate against the bacterial isolate (if plaques formed by the phage isolates were black or red). See Methods, revised "**Two-color plaque assay**".

The authors did not engage in the study of the temporal aspect of co-evolution, since they chose to take samples only after the end of the co-evolution rather than sample from the same location at different timepoints, which we agree would be highly ambitious and prone to several technical challenges. A discussion would be interesting, however.

Following the reviewer comment, we now discuss the aspect of temporal sampling of the coevolution assay in the discussion, see **lines 309-313**.

Sentence "... E. coli and T7 on swimming plates manifested ongoing coevolution for 15 days with symmetric dynamics, with neither species globally prevailing at any time point.": We suggest replacing the term symmetric by one reflecting the inherent asymmetry of the system.

We replaced symmetric with retaliating.

* Because of the number of free parameters, testing for epistasis is probably out of reach. Would it make sense, though, to see whether the epistatic effect between mutations in *opgG* and *waaJ* appears in the Lasso regression?

Unfortunately we could not search for epistasis between the *opgG* and *waaJ* mutations because we do not have an isolate with the *opgG* mutation alone in our dataset.

* A reference for MAGE and MASC-PCR might be helpful.

Indeed a reference was missing, references to both techniques were added.

* In the caption Supplementary Figure 5, is the reference to Supplementary Figure 5 intended?

We could not find the reference to Supplementary Figure 5 (now Supplementary Figure 10) mentioned above. However, we assured that there are no unintended references to figures. Also, we elaborated on the references in the figure caption: Supplementary Table 4 is referenced for the phage genotypes and supplementary table 5 is referenced for the phage phenotype, where these phage isolates show lower infectivity against bacterial isolates with an *opgG* mutation (**revised Figure 4, Supplementary Figure 10 and captions**) .

* In the caption of Supplementary Figure 6, do the numbers match up with the figure? Both plaques 1 and 2 appear green to us.

The numbers do match up with the figure but we accidentally omitted a reference to plaque number 3 in the caption, which might have caused a confusion (**now revised, Supplementary Figure 5**). In the left plate there are two phenotypes, one makes a bright green plaque (infects only the wildtype, #2), and the other makes a much darker plaque (infects both wildtype and evolved isolate, #1).

Reviewer #3 (Remarks to the Author):

Tamar and Kishony carried out experimental evolution of *E. coli* and T7 phage on four swim-agar plates that were imaged continuously and characterized bouts of coevolutionary dynamics as they occurred across space and time. They were able to observe multi-step coevolutionary cycles in distinct regions of space, with their own relatively uncoupled (?) dynamics [question mark here because actually I'm not sure this is something directly measured, but perhaps could be easily ascertained with the image data already on hand].

Tamar and Kishony further characterized infectivity between 116 phage isolates and 97 bacteria clones and identified canonical patterns observed in coevolutionary diversification. Further genetic analysis identified many mutations of possible adaptive benefit in both host and phage, but most interestingly mutations that seem to have reduced infectivity of the phage. Rather than clear arms-race dynamics, the phage evolved a host-switch, where adaptation to the new (T7wt) Resistant host genotype came at a cost to infectivity against the naive host.

This study will certainly be broadly interesting (and the movies are just stunning). The results are in line with observations seen in other bacteria-phage coevolution work, though it is rare to see all of these in a single study. The length and number of coevolutionary "bouts" is perhaps the most striking, which then enables the further analyses performed.

We thank the reviewer for the supportive review and the appreciation of our findings and for their helpful suggestions below.

I only have a few comments, none of which should prohibit publication.

1) The introduction reads quite abrupt and disconnected. Many times I'm left either wanting a sentence to connect the main points of the paragraph with the current study. For example, in paragraph 2 (L445 - 53), the introduction of a system of spatiotemporal tracking is introduced as a gap in the literature, rather than as a way of addressing a particular question.

Following the reviewer comment, we rephrased paragraph 2 so that the experimental system is now introduced as a way of addressing the question whether evolution in spatial environment may proceed with more adaptive steps and lead to higher degree of genetic co-diversification (see revised paragraph 2 and in particular lines 50-54)

Similarly, at the end of paragraph 3, I'm left wondering why coevolution experiments have failed to identify so many of the identified pathways to T7 resistance; a preview could really set the reader up for some of the interesting results to come!

We have now added a sentence at the end of paragraph 3 explaining the possible cause for the limited coevolution in well-mixed environments (see paragraph 3 lines 67-68).

2) [Minor] Line 38: The phrasing "subjected to lytic infection" read awkwardly to me, since we aren't the ones doing the infection per se.

We replaced "subjected to lytic infection" with "challenged by lytic infection".

3) [Minor] Line 41: I would eliminate the word "cycle", since reciprocal adaptation is itself a cycle

Indeed, we eliminated the word "cycle" from this sentence.

4) [Minor] Line 76: What is meant by "wide genetic variations"?

We rephrased the sentence by replacing "wide" with "multiple".

5) Lines 105-107: The median cycle being 2 is interesting, I wonder what the phenotypic outcome of these frequent 2-cycle bouts of coevolution are. Are they often the same set of adaptations in the 2-cycles, or are the endpoints of the 2-cycles found within longer cycles of coevolution and they're somehow cut short? What does the distribution of number of cycles look like?

Following the reviewer's questions, we now added a new figure showing the distribution of peaks (new Supplementary Figure 3). Based on the time-lapse movie of the initial coevolution we believe that although the initial expansion patterns in the four plates are similar, the initial population dynamics occur mostly at the center of the plates as opposed to later dynamics reaching the edges of the plates, suggesting that different areas represent different adaptation steps.

However, this cannot be tested directly due to sampling only at the end point (**see revised discussion, lines 309-313**).

6) L127-148: Since there are replicate infectivity analyses for the phage and bacteria wildtypes, I'd like to know how repeatable the automated scoring of infectivity was.

Following the reviewer's question we have added an analysis all replications of wildtype measurements (**see explanation and correlation results in lines 137-140**)

7) L165-193: One of the big differences between this study and other studies of host-parasite coevolution is that genotypes persist in the cultures without being removed. Does that mean that isolating spatial and temporal effects are more challenging? Or that dN/dS rates are potentially biased? If I'm right that the accumulation of previously fit genotypes that have since become disfavored remain in the culture and are sampled, I'd just like some mention of it.

This is indeed a fantastic comment. Previously fit genotypes may survive on the plate, even when less adapted to their local phage-bacteria population, although at lower rates and hence lower chances of being sampled. While this feature of coevolution on swimming plates indeed makes it more difficult to differentiate spatial and temporal effects, it also provides an advantage, given our single sampling point in time, of observing genotypes from earlier time points and to identify and analyze effects of accumulated mutations (such as *opgG* with *waaJ*). We anticipate that this survival of less fit genotypes may be reflected in the negative effect on phage infectivity we see in Figure 3b. We added an explanation with our observation (**Revised paragraph 9 lines 226-227**)

Reviewers' Comments:

Reviewer #1:

Remarks to the Author:

This MS has been modified by the authors, and all my points raised on the previous round have been addressed. This has made the MS easier to read, and increased the transparency of the sampling. Supplementary movie 3 (control experiment) is a great addition to the MS.

I have only a few notes.

Maybe combine the new paragraph (high throughput cross-infection assay for interaction mapping, line 122 onwards) with the next paragraph, because the actual analysis is in the next paragraph.

Supplementary figure 7 lacks explanation for WT, HS and HR.

Reviewer #2:

Remarks to the Author:

We - the report has been prepared collaboratively following journal guidelines - thank the authors and other reviewers for a stimulating scientific discussion.

The authors present a significantly revised manuscript which we recommend for publication following consideration of our remaining points and revision where applicable:

1. Code availability: Could the authors upload the code to Zenodo and/or GitHub instead of publishing it on the lab website?

2. In our previous report, we asked how the results differed from a bacteria-phage co-evolution experiment in liquid.

The authors have presented the technical challenges in creating equivalent conditions between a liquid culture and their spatially structured system and how the differing speeds of host growth in the two systems may lead to differing outcomes. We were aware of these difficulties and appreciate the detailed discussion in the manuscript.

However, we aimed to see better support for the claim that evolution in a spatially structured environment leads to different outcomes. For example: How different are the bacterial mutants evolved from the experiments in this manuscript to previously reported liquid experiment outcomes, in terms of genotype / phenotype? The manuscript is quite general on this point and a more detailed discussion of the related literature which is already cited (probably Refs. 12, 13, 15, 21) would be helpful.

3. We have a few remaining questions on the role of mlaA and gmhA:

3a. The authors report the novel role of mlaA. They demonstrate by genetic reconstruction how the mlaA:D,if mutation leads to increased resistance. However, Fig. 3a also suggests that the mutation mlaA:IS leads to increased sensitivity of the bacteria to the phage T7. Can the authors discuss the differences in outcome between the two mutations? We feel it is an interesting observation that should be highlighted.

3b. The authors clarify that gmhA:IS is actually ghmA:IS + mlaA:XXX. Is this the same mutation in mlaA reported elsewhere? In any case, should this not be reflected in Fig. 3?

3c. Moreover, the arguments presented in the reply to referees and the updated main text regarding mlaA and gmhA cancelling each other's effect left us further confused. This might be due to the fact that it is unclear from the text which mutation in mlaA is meant. In any case, more detail would be helpful given that mutations in mlaA can lead to both resistance and sensitivity.

3d. It is unclear how gmhA(+mlaA) leading to more sensitivity than other mlaA mutations can also lead to "more resistant" compared to the ancestor strain", given the positive association shown in Fig. 3 (assuming 0 to correspond to the ancestor).

4. Classification is a bit glanced over in the manuscript, i.e., there is no reference to Methods in the main text and it is still not clear to us what counts as a phenotypic class in Figure 2. It would be good to see these points addressed.

5. Minor point on Fig. 1: Is there a formal criterion for the classification into what constitutes a highly dynamic area? It is of course fine if it is determined 'by eye' for illustration purposes.

6. We suggest rewording "In contrast with the bacterial mutations, phage mutations included mutations associated both negatively and positively with infectivity" to "In contrast with the bacterial mutations, phage mutations included many more mutations associated negatively with infectivity" to better reflect Fig. 3.

7. It would be useful to reference relevant Methods section titles throughout given the breadth of methods applied in this manuscript.

8. "The robustness of the infectivity score was tested via measurement of correlation between all three bacterial wildtype infection scores and all four phage wildtype infection scores which showed a good agreement": An additional figure would be helpful.

9. We now better understand the report around CFP / YFP markers. However, we are left wondering whether "the information of marker ratio was thereby not utilized" means that analysis was done blind to the marker or only YFP colonies were selected.

10. "5µl of the mixed culture ($6 \cdot 10^6 \pm 1 \cdot 10^6$ MG1655-YFP cells and 13 ± 2 T7 virions) was inoculated at the center of each plate." Should this read a mixture of CFP and YFP?

11. Methods on 'Sampling': Could the authors provide full detail on 'phosphate buffered saline'?

12. Methods on 'Single bacterial colonies isolation': It is not clear to us whether isolation happens from frozen stocks or directly after sampling. In this context, the meaning of the sentence 'Stocks ... were maintained at -80C' in the middle of the description is unclear.

Reviewer #3:

Remarks to the Author:

The authors have carefully gone through the reviewers comments, and have addressed my concerns fully. The additions in the paper, especially the newly added methods section, greatly improve the manuscript. The included caveats in the discussion clearly delimit the scope of this study as well as why certain experiments are intractable or at least improperly comparable. I think this work is ready for publication after only minor edits.

Minor comments:

Line 46: subpopulation -> subpopulations

Lines 49-54: There is a bit of strange language here, where the subject of the sentence seems to change between adapting populations or one of the particular players, or the experimenters doing the study. I suggest spending a bit of time editing this section. Also, I think you may mean uncover instead of unravel.

Reviewer #4:
None

Point-by-point response to reviewer comments

Reviewer #1 (Remarks to the Author):

This MS has been modified by the authors, and all my points raised on the previous round have been addressed. This has made the MS easier to read, and increased the transparency of the sampling. Supplementary movie 3 (control experiment) is a great addition to the MS.

We thank the reviewer for the thoughtful review and constructive comments which helped improve the transparency and clarity of the manuscript.

I have only a few notes.

Maybe combine the new paragraph (high throughput cross-infection assay for interaction mapping, line 122 onwards) with the next paragraph, because the actual analysis is in the next paragraph.

Following the reviewer's comment we combined both paragraphs into one, under the title of the originally second paragraph.

Supplementary figure 7 lacks explanation for WT, HS and HR.

Thank you, missing labels were added.

Reviewer #2 (Remarks to the Author):

We - the report has been prepared collaboratively following journal guidelines - thank the authors and other reviewers for a stimulating scientific discussion.

The authors present a significantly revised manuscript which we recommend for publication following consideration of our remaining points and revision where applicable:

We thank the reviewers for the thorough review and the highly constructive and insightful comments which helped improve the manuscript.

1. Code availability: Could the authors upload the code to Zenodo and/or GitHub instead of publishing it on the lab website?

Following the reviewers request, the code is now uploaded to GitHub and the raw data is uploaded to Zenodo.

2. In our previous report, we asked how the results differed from a bacteria-phage co-evolution experiment in liquid.

The authors have presented the technical challenges in creating equivalent conditions between a liquid culture and their spatially structured system and how the differing speeds of host growth in the two systems may lead to differing outcomes. We were aware of these difficulties and appreciate the detailed discussion in the manuscript.

However, we aimed to see better support for the claim that evolution in a spatially structured environment leads to different outcomes. For example: How different are the bacterial mutants evolved from the experiments in this manuscript to previously reported liquid experiment outcomes, in terms of genotype / phenotype? The manuscript is quite general on this point and a more detailed discussion of the related literature which is already cited (probably Refs. 12, 13, 15, 21) would be helpful.

Following the reviewer's comment, we now describe in detail the phenotypic and genotypic outcomes of previous coevolution experiments in liquid environments (**Discussion, lines 276-285**).

3. We have a few remaining questions on the role of mlaA and gmhA:

3a. The authors report the novel role of mlaA. They demonstrate by genetic reconstruction how the mlaA:D,if mutation leads to increased resistance. However, Fig. 3a also suggests that the mutation mlaA:IS leads to increased sensitivity of the bacteria to the phage T7. Can the authors discuss the differences in outcome between the two mutations? We feel it is an interesting observation that should be highlighted.

This is indeed an interesting observation. We now explain the two types of mutations and their expected phenotypes in more detail (**lines 195-204**).

3b. The authors clarify that gmhA:IS is actually ghmA:IS + mlaA:XXX. Is this the same mutation in mlaA reported elsewhere? In any case, should this not be reflected in Fig. 3?

While the isolate carrying the gmhA:IS mutation also carries an in-frame insertion in mlaA, the positive coefficient is assigned solely to the mutation gmhA:IS. Following this question we now clarify this in a more detailed explanation of the gmhA mutation and its positive-infectivity-associated coefficient (**lines 204-212**).

3c. Moreover, the arguments presented in the reply to referees and the updated main text regarding mlaA and gmhA cancelling each other's effect left us further confused. This might be due to the fact that it is unclear from the text which mutation in mlaA is meant. In any case, more detail would be helpful given that mutations in mlaA can lead to both resistance and sensitivity.

Following comments 3a-c we now explain the argument in more detail and with a better reference to the relevant mutations (**lines 195-212**).

3d. It is unclear how *gmhA*(+*m1aA*) leading to more sensitivity than other *m1aA* mutations can also lead to “more resistanc[ce] compared to the ancestor strain”, given the positive association shown in Fig. 3 (assuming 0 to correspond to the ancestor).

Since the bars in Figure 3 represent the coefficients assigned by the model to each mutation (with 0 corresponding to a mutation with no effect on infectivity), we deduce that the contribution of the *m1aA* in-frame mutations to resistance is comparable to the contribution of the *gmhA* mutation to sensitivity. We hope that now this is explained better in the text, following the changes related to the three points above.

4. Classification is a bit glanced over in the manuscript, i.e., there is no reference to Methods in the main text and it is still not clear to us what counts as a phenotypic class in Figure 2. It would be good to see these points addressed.

We now chose a more rigorous threshold for differentiating phenotypic classes which we define as 40% of the maximal linkage (**line 155 and revised Methods: Dendrogram construction**)

5. Minor point on Fig. 1: Is there a formal criterion for the classification into what constitutes a highly dynamic area? It is of course fine if it is determined ‘by eye’ for illustration purposes.

We chose to display areas with the maximal, or almost the maximal number of peaks in the corresponding replicate. A second criteria was that neighboring pixels showed a similar number of peaks.

6. We suggest rewording “In contrast with the bacterial mutations, phage mutations included mutations associated both negatively and positively with infectivity” to “In contrast with the bacterial mutations, phage mutations included many more mutations associated negatively with infectivity” to better reflect Fig. 3.

As the reviewers suggested, the sentence was rewarded.

7. It would be useful to reference relevant Methods section titles throughout given the breadth of methods applied in this manuscript.

As the reviewers suggested, all referenced to Methods now include the name of the relevant method section.

8. “The robustness of the infectivity score was tested via measurement of correlation between all three bacterial wildtype infection scores and all four phage wildtype infection scores which showed a good agreement”: An additional figure would be helpful.

Following the reviewer’s comment, figures showing the correlation between infectivity scores of different wildtype replicates were added (**new Supplementary Figure 6c-d**).

9. We now better understand the report around CFP / YFP markers. However, we are left wondering whether “the information of marker ratio was thereby not utilized” means that analysis was done blind to the marker or only YFP colonies were selected.

Only YFP colonies were selected for further analysis (less than 7% of isolated colonies contained CFP markers). We now clarify this in the text (**revised Methods, Coevolution experiment**).

10. “5µl of the mixed culture ($6 \cdot 10^6 \pm 1 \cdot 10^6$ MG1655-YFP cells and 13 ± 2 T7 virions) was inoculated at the center of each plate.” Should this read a mixture of CFP and YFP?

CFP cell count was previously omitted, we now added it for clarity (**revised Methods, Coevolution experiment**).

11. Methods on ‘Sampling’: Could the authors provide full detail on ‘phosphate buffered saline’?

Full product and preparation information was added.

12. Methods on ‘Single bacterial colonies isolation’: It is not clear to us whether isolation happens from frozen stocks or directly after sampling. In this context, the meaning of the sentence ‘Stocks ... were maintained at -80C’ in the middle of the description is unclear.

Isolation happened from thawed frozen stocks, as we now clearly state in the text (**revised Methods, Single bacterial colonies isolation**).

Reviewer #3 (Remarks to the Author):

The authors have carefully gone through the reviewers comments, and have addressed my concerns fully. The additions in the paper, especially the newly added methods section, greatly improve the manuscript. The included caveats in the discussion clearly delimit the scope of this study as well as why certain experiments are intractable or at least improperly comparable. I think this work is ready for publication after only minor edits.

We thank the reviewer for the positive and careful review and are grateful for the valuable comments and suggestions which strengthened the coherence of the manuscript.

Minor comments:

Line 46: subpopulation -> subpopulations

Thank you, the typo was fixed.

Lines 49-54: There is a bit of strange language here, where the subject of the sentence seems to change between adapting populations or one of the particular players, or the experimenters doing the study. I suggest spending a bit of time editing this section.

Following the reviewer's comment, we reworded this section.

Also, I think you may mean uncover instead of unravel.

Thank you, we changed "unravel" to "uncover".